



Total organic carbon and contribution from speciated organics in cloud water: Airborne data
analysis from the CAMP[2]Ex field campaign
Connor Stahl[1], Ewan Crosbie[2,3], Paola Angela Bañaga[4,5], Grace Betito[4], Rachel A. Braun[1], Zenn
Marie Cainglet[4,5], Maria Obiminda Cambaliza[4,5], Melliza Templonuevo Cruz[4,6], Julie Mae
Dado[7], Miguel Ricardo A. Hilario[4,8], Gabrielle Frances Leung[4,9], Alexander B. MacDonald[1],
Angela Monina Magnaye[7], Jeffrey Reid[10], Claire Robinson[2,3], Michael A. Shook[2], James
Bernard Simpas[4,5], Shane Marie Visaga[5,7], Edward Winstead[2,3], Luke Ziemba[2], Armin
Sorooshian[1,8]
[1]Department of Chemical and Environmental Engineering, University of Arizona, Tucson,
Arizona, 85721, USA
[2]NASA Langley Research Center, Hampton, Virginia, 23666, USA
[3]Science Systems and Applications, Inc., Hampton, Virginia, 23666, USA
[4]Air Quality Dynamics-Instrumentation & Technology Development Laboratory, Manila
Observatory, Quezon City, 1108, Philippines
[5]Department of Physics, School of Science and Engineering, Ateneo de Manila University,
Quezon City, 1108, Philippines
[6]Institute of Environmental Science and Meteorology, University of the Philippines, Diliman,
Quezon City, 1101, Philippines
[7]Regional Climate Systems Laboratory, Manila Observatory, Quezon City, 1108, Philippines
[8]Department of Hydrology and Atmospheric Sciences, University of Arizona, Tucson, Arizona,
85721, USA
[9]Department of Atmospheric Science, Colorado State University, Fort Collins, Colorado 80521,
USA
[10]Marine Meteorology Division, Naval Research Laboratory, Monterey, California 93943, USA
*Correspondence to: armin@email.arizona.edu*





**Abstract**
This work focuses on total organic carbon (TOC) and contributing species in cloud water over
Southeast Asia using a rare airborne dataset collected during NASA's Cloud, Aerosol and
Monsoon Processes Philippines Experiment (CAMP$^2$Ex), in which a wide variety of maritime
clouds were studied, including cumulus congestus, altocumulus, altostratus, and cumulus.
Knowledge of TOC levels and their contributing species is needed for improved modeling of
cloud processing of organics and to understand how aerosols and gases impact and are impacted
by clouds. This work relies on 159 samples collected with an Axial Cyclone Cloud water
Collector at altitudes of 0.2 – 6.8 km that had sufficient volume for both TOC and speciated
organic composition analysis. Species included monocarboxylic acids (glycolate, acetate,
formate, and pyruvate), dicarboxylic acids (glutarate, adipate, succinate, maleate, and oxalate),
methanesulfonate (MSA), and dimethylamine (DMA). TOC values range between 0.018 –
13.660 ppm C with a mean of 0.902 ppm C. The highest TOC values are observed below 2 km
with a general reduction aloft. An exception is samples impacted by biomass burning for which
TOC remains enhanced as high as 6.5 km (7.048 ppm C). Estimated total organic matter derived
from TOC contributes a mean of 30.7% to total measured mass (inorganics + organics).
Speciated organics contribute (on carbon mass basis) an average of 30.0% to TOC in the study
region, and account for an average of 10.3% to total measured mass.
The order of the average contribution of species to TOC, in decreasing contribution of carbon
mass, is as follows: acetate (14.7 ± 20.5%), formate (5.4 ± 9.3%), oxalate (2.8 ± 4.3%), DMA
(1.7 ± 6.3%), succinate (1.6 ± 2.4%), pyruvate (1.3 ± 4.5%), glycolate (1.3 ± 3.7%), adipate (1.0
± 3.6%), MSA (0.1 ± 0.1%), glutarate (0.1 ± 0.2%), maleate (< 0.1 ± 0.1%). Approximately 70%
of TOC remains unaccounted for, thus highlighting the complex nature of organics in the study
region; samples collected in biomass burning plumes have up to 95.6% of unaccounted TOC
mass based on the species detected. Consistent with other regions, monocarboxylic acids
dominate the speciated organic mass (~75%) and are about four times in greater abundance than
dicarboxylic acids.
Samples are categorized into four cases based on back-trajectory history revealing source-
independent similarity between the bulk contributions of monocarboxylic and dicarboxylic acids
to TOC (16.03% – 23.66% and 3.70% – 8.75%, respectively). Furthermore, acetate, formate,
succinate, glutarate, pyruvate, oxalate, and MSA are especially enhanced during biomass burning
periods, attributed to peat emissions transported from Sumatra and Borneo. Lastly, dust ($Ca^{2+}$)
and sea salt ($Na^+$/$Cl^-$) tracers exhibit strong correlations with speciated organics, thus supporting
how coarse aerosol surfaces interact with these water-soluble organics.





## 1. Introduction


The last two decades witnessed an acceleration of research to unravel the nature of the organic
fraction of airborne particles, including speciation (Hallquist et al., 2009; Kanakidou et al.,
2005), with implications for how particles impact air quality, public health, and the planet's
energy balance. However, there has been much less progress on organic research for cloud
droplets, owing largely to the inaccessibility of clouds as compared to particles that can be
measured more easily near the surface. Analyzing organic matter in cloud water will lead to
better understanding of secondary aerosol formation and the nature of cloud condensation nuclei
(CCN) that form droplets. The interaction of aerosol particles and clouds interact constitutes the
largest uncertainty in estimating total anthropogenic radiative forcing (IPCC, 2013), which
motivates using cloud composition as a tool to learn about these processes (MacDonald et al.,
2020). Characterizing cloud water composition is insightful for atmospheric chemical processes
such as the removal of gases that would otherwise participate in gas-phase reactions and for
aqueous reactions that yield products without an efficient gas-phase source (e.g., dicarboxylic
acids) (Ervens et al., 2013). While modeling of sulfate production in clouds is fairly mature
(Barth et al., 2000; Faloona, 2009; Kreidenweis et al., 2003; Liu et al., 2021), the formation and
evolution of organics in cloud water is much more poorly constrained (Ervens, 2015).
Advancing this research requires in situ measurements of cloud water composition. Among the
most common methods of characterizing the organic fraction of cloud water samples is total
organic carbon (TOC) analysis. Whether it is cloud water or fog water, most studies have shown
that (i) TOC is enhanced in air masses with higher anthropogenic influence (Collett Jr. et al.,
1998; Deguillaume et al., 2014; Herckes et al., 2013; Raja et al., 2009); (ii) ~40% – 85% of the
TOC is attributed to unidentified species (Benedict et al., 2012; Boris et al., 2016; Boris et al.,
2018; Herckes et al., 2002; Raja et al., 2008); (iii) organic acids usually account for ≲15% of the
TOC (Deguillaume et al., 2014; Gioda et al., 2011; Straub et al., 2007); (iv) monocarboxylic
acids are more abundant than dicarboxylic acids (Löflund et al., 2002); and (v) acetic and formic
acids are the most dominant organic acids contributing to TOC (Collett Jr. et al., 2008; Gioda et
al., 2011). Most of the aforementioned studies focused on fog, therefore motivating a closer look
at cloud water, as solute concentrations depend on the type of aqueous medium (Fig. 1). More
specifically, TOC concentrations are reported to be higher in fog water relative to rain water
(Kim et al., 2020), while cloud water solute concentrations exceed those in rain water (Decesari
et al., 2005; Gioda et al., 2008).
Southeast Asia is an ideal laboratory to investigate the nature of TOC and its constituents as it is
impacted by a multitude of emissions sources in an environment with persistent cloud cover from
a variety of cloud types (e.g., shallow cumulus and cumulus congestus clouds) (Reid et al.,
2013). The complex meteorology of the region makes it very difficult to model (Wang et al.,
2013; Xian et al., 2013), but simultaneously provides a remarkable opportunity to learn more
about how aerosols impact (and are impacted by) tropical cloud systems. A knowledge gap exists
as there have been no studies of cloud composition in this region based on airborne
measurements. Analysis of fog water at Baengnyeong Island in the eastern Yellow Sea revealed
that organic acids accounted for 36 – 69% of TOC (Boris et al., 2016). The Acid Deposition





Monitoring Network in East Asia (https://www.eanet.asia/) provides data on wet deposition at
surface sites such as at the Manila Observatory (Metro Manila, Philippines) (Ma et al., 2021) and
is limited to inorganic ions. Previous studies such as the Seven South East Asian Studies
(7SEAS) (Reid et al., 2013) and the Cloud, Aerosol and Monsoon Processes Philippines
Experiment (CAMP²Ex) weatHEr and CompoSition Monitoring (CHECSM) were carried out in
this region; however, these campaigns were ground and ship-based, and focused mainly on
aerosol particles and not cloud composition (Hilario et al., 2020b; Reid et al., 2015; Reid et al.,
109 2016).

Recent studies in Metro Manila, Philippines provide the following results of relevance to this
work: (i) a third to a half of the total aerosol particle mass is often unaccounted for after
considering water-soluble species (inorganic and organic acid ions and elements) and black
carbon (Cruz et al., 2019; Stahl et al., 2020); (ii) organic acids account for less than 1% of total
aerosol mass, with oxalate being the most abundant acid (Stahl et al., 2020); (iii) organic acid
levels are more enhanced during biomass burning periods (Hilario et al., 2020a), especially
succinate and oxalate (Braun et al., 2020; Stahl et al., 2020); and (iv) wet deposition samples
clearly show the influence of biomass burning tracer species on cloud composition (Ma et al.,
2021). Based on these points, we test two hypotheses: (i) the relative contribution of organic
acids to TOC will exceed what was observed at the surface layer over Metro Manila owing to
more aged air masses aloft as compared to the surface layer in Metro Manila exposed to fresher
emissions; and (ii) clouds impacted by biomass burning emissions will exhibit chemical profiles
shifted to higher TOC levels and with a greater portion of that TOC accounted for by organic
acids. To address these hypotheses in addition to characterizing the organic fraction of cloud
water, we utilized a rich set of cloud water samples collected around the Philippines during
CAMP²Ex between August and October in 2019. The subsequent results and discussion focus on
TOC concentrations in addition to the relative contribution and interrelationships between a suite
of organic species (organic acids, methanesulfonate, dimethylamine) spatially, and as a function
of altitude and air mass source origin. A unique aspect of this dataset is the large sample number
with both TOC and speciated organic acid information from an airborne platform.

**2. Methods**
**2.1 Study overview**

A total of 159 cloud water samples were collected on the NASA P-3B Orion aircraft across 19
research flights (RF; 23 August – 5 October 2019) during CAMP²Ex that were measured for
ions, pH, and TOC. Flights were based out of Clark International Airport (15.189°N, 120.547°E)
and extended to regions around the island of Luzon (Fig. 2). Cloud water samples were collected
over a wide range of altitudes ranging from 0.2 – 6.8 km.

**2.2 Cloud water collection and handling**





Samples were collected using the Axial Cyclone Cloud water Collector (AC3), (Crosbie et al., 2018), which efficiently collects cloud droplets with effective diameters > 20 µm. The AC3 has a shutter attached to a servo motor allowing the collector to be closed when not in a cloud to prevent contamination. Samples were collected between 10 seconds and 10 minutes depending on cloud availability and liquid water content (i.e., shorter times possible with higher liquid water content). Cloud water was collected in prewashed 15-mL plastic conical vials. Due to thorough prewashing of the plastic conical vials, leaching of organics into samples was negligible. Before each flight, the collector was flushed with ~ 1 L of ultra-purified Milli-Q water (18.2 MΩ-cm) prior to obtaining two blank samples. Blanks were also collected post-flight that were similarly flushed prior to being collected. During flight, samples were collected and stored in a cooler with sufficient ice packs to reduce possible decomposition. After flights, samples were immediately taken to an onsite laboratory where sample volumes were recorded and analyzed for ionic composition, TOC, and pH. A background was subtracted from the samples based on the 10[th] percentile of all samples and blanks collected during the campaign. Excess samples were stored in a refrigerator for future analyses that are outside the scope of this study.

### 2.3 Cloud water analysis
### 2.3.1 Ion chromatography

Cloud water was speciated using ion chromatography (IC; Dionex ICS-2100) immediately after each flight to reduce the possibility of degradation of the samples. Measured anionic species of interest were glycolate, acetate, formate, methanesulfonate, pyruvate, glutarate, adipate, succinate, maleate, oxalate, $Cl^-$, $NO_2^-$, $Br^-$, $NO_3^-$, and $SO_4^{2-}$. Measured cations were $Na^+$, $NH_4^+$, $K^+$, dimethylamine (DMA), $Mg^{2+}$, and $Ca^{2+}$. A 23-minute instrument method was used for both anion and cation columns with a 2-minute equilibration period, yielding a 25-minute sampling period per sample. The instrument flow rate was 0.4 mL min$^{-1}$. The anions were measured using a Dionex IonPac AS11-HC 2 × 250 mm column, a Dionex AERS 500e suppressor, and with potassium hydroxide as the eluent. The cations were measured using a Dionex IonPac CS12A 2 × 250 mm column, a Dionex CERS 500e suppressor, and using methanesulfonic acid (MSA) as the eluent. The instrument methods used for analysis are as follows: (i) for anions the eluent concentration started at 1 mM, ramped up to 4 mM between 0 – 10 minutes, ramped up to 6 mM between 10 – 11 minutes, and finally ramped up to 7 mM between 11 – 23 minutes using a suppressor current of 8 mA; (ii) for cations the eluent concentration started at 5 mM and remained isocratic from 0 – 10 minutes, ramped up to 18 mM between 10 – 12 minutes, and finally remained isocratic at 18 mM from 12 – 23 minutes using a suppressor current of 22 mA. The limits of detection (LOD) for these species can be found in Table 1.

### 2.3.2 Total organic carbon

Total organic carbon (TOC) was measured using a Sievers 800 Turbo TOC analyzer. Sample aliquots were diluted to obtain the minimum volume needed by the instrument. The TOC





analyzer was operated in turbo mode and TOC values were averaged over a stable concentration
period. Milli-Q water was used as an internal reference and calibrations were performed before
and after each batch of samples was analyzed (i.e., one batch every ~3 – 4 flights) using a range
of different concentrations from an oxalate standard solution. A volume of approximately 10 mL
was used for each measurement and MQ water was used intermittently to flush the instrument
between each sample.

**2.3.3 Units**
While many studies report concentrations in terms of air-equivalent concentrations, we instead
use the native liquid-phase concentrations. Aqueous concentrations of TOC and individual
molecular components are reported in units of ppb (i.e., parts per billion by mass). TOC
concentrations are specific to the mass of carbon atoms only, while molecules measured by IC
correspond to the specific mass of the species (unless noted otherwise). TOC was converted to
total organic matter (TOM) via multiplication by 1.8 (Zhang et al., 2005).
The choice to focus on aqueous- rather than air-equivalent concentrations was made for various
reasons. First, our analysis focuses heavily on relative amounts of species that were unaffected
by multiplying native aqueous units by cloud liquid water content. Second, the definition of
liquid water content applied by studies can vary widely based on the lower and upper bound of
what is considered a droplet. Third, relationships between solute concentrations in cloud water
and liquid water content, anticipated from nucleation scavenging, are ineffective when gases like
acetic and formic acids adsorb directly to droplets rather than having been part of the initial CCN
activating into droplets (Khare et al., 1999; Marinoni et al., 2004). Lastly, many studies of cloud
water composition that our results can be contrasted with also use liquid units. The primary
liquid units reported for cloud water concentrations are ppm and ppb.

**2.4 Aerosol Composition**
To complement the cloud water composition results, we use aerosol composition results from the
High-Resolution Time-of-Flight Aerosol Mass Spectrometer (AMS; Aerodyne, Inc.), which
reports non-refractory composition for the submicrometer range (DeCarlo et al., 2006). As
summarized by Hilario et al. (2021), the AMS deployed in CAMP²Ex functioned in 1 Hz Fast-
MS mode with data averaged to 30 s time resolution with the lower limit of detection (units of
$\mu g\ m^{-3}$) as follows for the measured species: organic (0.169), $NH_4^+$ (0.169), $SO_4^{2-}$ (0.039), $NO_3^-$
(0.035), $Cl^-$ (0.036). Negative mass concentrations were recorded owing to the difference method
used with the limits of detection. These negative values were included in the analyses to avoid
positive biases but were interpreted as zero concentrations. We also use data specifically for the
mass spectral marker representative of acid-like oxygenated organic species ($m/z$ 44=$COO^+$)
(Aiken et al., 2008) and its mass relative to total organic mass ($f_{44}$). AMS data were omitted from
analysis if total mass of all detected species was $< 0.5\ \mu g\ m^{-3}$. By convention for airborne
sampling, AMS data are reported at standard temperature and pressure (STP; 273 K, 1013 hPa).



AMS data were reported separately for cloud-free and cloudy conditions owing to the use of a
counterflow virtual impactor (CVI) inlet (Brechtel Manufacturing Inc.) (Shingler et al., 2012) in
clouds to isolate and dry droplets, leaving the residual particles for sampling by the AMS. Cloud-
free data involve sampling with a separate inlet designed by the University of Hawaii
(McNaughton et al., 2007). For cloud-free AMS results, data were selected 60 seconds before
and after each cloud water sample's start and end time, respectively, when the aircraft was not in
cloud. CVI-AMS data were reported for data collected within the period of cloud water
collection. It should be noted that cloud-free AMS data are missing for some cloud water
samples when the CVI was still in use for the 60 s before and after a sample's start and end time,
respectively.

**2.5 HYSPLIT**
Air mass origination was determined using 5-day back trajectories from the National Oceanic
and Atmospheric Administration (NOAA) Hybrid Single Particle Lagrangian Integrated
Trajectory model (HYSPLIT) (Rolph et al., 2017; Stein et al., 2015). Trajectories were generated
at 1-minute temporal resolution with meteorological inputs from the Global Forecast System
(GFS) reanalysis with a horizontal resolution of $0.25° \times 0.25°$ using the "model vertical velocity"
method.

**2.6 NAAPS**
The Navy Aerosol Analysis and Prediction System (NAAPS) global aerosol model was
implemented to assist in identifying biomass burning cases (Lynch et al., 2016)
(https://www.nrlmry.navy.mil/aerosol/). NAAPS relies on global meteorological fields from the
Navy Global Environmental Model (NAVGEM) (Hogan and Brody, 1993; Hogan and Rosmond,
1991) that analyzes and forecasts a $1°\times1°$ grid with 6-hour intervals with 24 vertical levels. In
terms of identifying biomass burning cases, surface smoke concentrations were examined.

**3. Cumulative Results**
**3.1 Concentration Statistics**
TOC values ranged from 0.018 – 13.660 ppm C, with median and mean levels of 0.546 and
0.902 ppm C, respectively (Table 1). Samples in this study exhibited nearly the lowest mean
TOC value of all cloud water studies surveyed in Fig. 1, with the other lowest values being in
San Diego, California (0.85 ppm C), (Straub et al., 2007) and East Peak, Puerto Rico (0.90 ppm
C), (Gioda et al., 2008; Gioda et al., 2011; Reyes-Rodríguez et al., 2009). The CAMP²Ex dataset
exhibited the lowest minimum TOC value of all shown studies. For context, the highest mean
and maximum TOC levels in cloud water studies were 34.5 and 51.7 ppm C, respectively, at Jeju
Island, Korea, while the peak dissolved organic carbon (DOC) level in cloud water was 85.6 ppm
C at Mt. Tai, China. For comparisons to published cloud water measurements, DOC and TOC
are assumed to be sufficiently similar in nature to directly compare values. Differences in TOC





between our study and others can partly be attributed to the different types of clouds studied in
the CAMP²Ex region (e.g., cumulus congestus, cumulus, altocumulus, altostratus) and the higher
collection altitudes being conducive to enhanced liquid water contents and droplet sizes than
stratocumulus clouds in regions like the northeastern (Straub et al., 2007) and southeastern
Pacific Ocean (Benedict et al., 2012). Previous studies have primarily sampled stratocumulus or
stratus clouds (Fig. 1). Also, some of our samples may have included rain water, which naturally
has lower levels of TOC than cloud water due to dilution (Fig. 1). To illustrate the importance of
this dilution effect, an average of the mean values from the Fig. 1 studies shows the following
(ppm C): Fog = 17.8, cloud = 6.4, rain = 0.6. We further note that direct comparisons of our
results to others need to factor that water collectors have different transmission efficiency
behavior as a function of droplet size.
The order of species is as follows in terms of decreasing average contribution of C mass relative
to total TOC: acetate ($14.7 \pm 20.5\%$), formate ($5.4 \pm 9.3\%$), oxalate ($2.8 \pm 4.3\%$), DMA ($1.7 \pm$
$6.3\%$), succinate ($1.6 \pm 2.4\%$), pyruvate ($1.3 \pm 4.5\%$), glycolate ($1.3 \pm 3.7\%$), adipate ($1.0 \pm$
$3.6\%$), MSA ($0.1 \pm 0.1\%$), glutarate ($0.1 \pm 0.2\%$), and maleate ($< 0.1 \pm 0.1\%$). An average of
70.0% of TOC mass went unaccounted for. The predominant sources and production pathways
of these species are briefly explained here. Precursor emissions sources for acetate and formate
include plants, soil, vehicles, and biomass burning, with key production routes including
oxidation of isoprene, ozonolysis of olefins, and peroxy radical reactions (Khare et al., 1999, and
references therein). Pyruvate is considered the most abundant aqueous reaction product of
methylglyoxal, generated by the oxidation of gas-phase anthropogenic volatile organic
compounds (Boris et al., 2014; Carlton et al., 2006; Lim et al., 2013; Stefan et al., 1996; Tan et
al., 2010). Glycolate has been linked to aqueous processing of acetate and a precursor for
glyoxylate (Boris et al., 2014) and formed via oxidation of glycolaldehyde by hydroxide radicals
(Thomas et al., 2016). Oxalate is the most abundant dicarboxylic acid across different
environments (Cruz et al., 2019; Stahl et al., 2020; Yang et al., 2014; Ziemba et al., 2011) and
can be emitted directly by biogenic sources (Boone et al., 2015; Kawamura and Kaplan, 1987),
combustion exhaust (Kawamura and Kaplan, 1987; Kawamura and Yasui, 2005), and biomass
burning (Narukawa et al., 1999; Yang et al., 2014); however, it is also formed through the
oxidation and degradation of longer chain organic acids and acts as a notable tracer for cloud
processing (Ervens et al., 2004; Sorooshian et al., 2006). Saturated organics like glutarate,
adipate, and succinate are linked to fresh emissions and mainly from ozonolysis of cyclic alkenes
(such as from vehicular emissions) in the study region (Hatakeyama et al., 1985; Stahl et al.,
2020). Maleate can be secondarily formed from the photooxidation of benzene (Rogge et al.,
1993) or from the primary emissions of combustion engines (Kawamura and Kaplan, 1987).
Alkyl amines (i.e., DMA) have numerous sources such as biomass burning, vehicular emissions,
industrial activity, animal husbandry, waste treatment, and the ocean (Youn et al., 2015). Finally,
MSA is formed via photooxidation reactions involving dimethylsulfide (DMS) from oceanic
emissions (Berresheim, 1987; Saltzman et al., 1983) or dimethyl sulfoxide (DMSO) from
anthropogenic emissions (Yuan et al., 2004), in addition to being linked to agricultural emissions
and biomass burning (Sorooshian et al., 2015).



Measured organic species were further grouped into categories: monocarboxylic acids (MCA;
glycolate, acetate, formate, pyruvate), dicarboxylic acids (DCA; glutarate, adipate, succinate,
maleate, oxalate), and measured organics (MO = sum of MCA, DCA, MSA, DMA). Total MCA
concentrations accounted on average for ~75% of MO and were approximately four times larger
than those of DCAs. MO values ranged from 29.5 – 10815.3 ppb, accounting for an average of
30.0% (median 23.8%) of TOC when masses were converted to just the C masses of the
measured species (Table 1). Examples of other undetected organics include tricarboxylic acids,
aromatics, alcohols, sugars, carbohydrates, and aldehydes. Previous studies reported undetected
species accounting for ~45% (Boris et al., 2016) and 82.9% (Boris et al., 2018) of organics.
Interestingly, the ionic charge balance for the 159 samples shows a slight cation deficit (Fig. S1),
with a slope of 1.04 (i.e., anion charge on y-axis). This fairly good charge balance suggests that
detected organic species were balanced by cations detected via IC analysis. Species contributing
to the slight cation deficit likely include metal cations and $H^+$.
TOC was converted to total organic matter (TOM) by multiplying it by 1.8 (Zhang et al., 2005),
as in other cloud water studies (Boris et al., 2016; Boris et al., 2018), in order to compare it to
total measured mass (i.e., sum of TOM, Na, $NH_4^+$, $K^+$, $Mg^{2+}$, $Ca^{2+}$, $Cl^-$, $NO_2^-$, $Br^-$, $NO_3^-$, $SO_4^{2-}$).
We caution that using a fixed 1.8 conversion value yields uncertainty as samples were collected
in a range of air masses, but 1.8 is a value fairly intermediate to those reported in the literature:
$1.6 \pm 0.2$ for urban aerosols (Turpin and Lim, 2001), $2.07 \pm 0.05$ in nonurban areas (Yao et al.,
2016), and values for biomass burning organic aerosols ranging from 1.56 – 2.0 (Aiken et al.,
2008; Turpin and Lim, 2001) based on fuel type and combustion condition (Aiken et al., 2008).
Higher values are expected for more oxidized organics. Estimated TOM accounted for a median
and mean of 23.2% and 30.7%, respectively, of total measured mass, with the maximum for a
single sample being 95.1%. The median and mean ratios of MO to TOM were 38.1% and 46.4%,
respectively. Furthermore, the median and mean ratios of MO to total measured mass were 7.2%
and 10.3%, respectively, with a maximum of 57.6%. On average, chloride, sulfate, and nitrate
were the most abundant species ($\geq 12.6\%$), with the median and mean ratio of total inorganic
mass to TOM being 3.3 and 5.8, respectively.
Our calculated percentages of MO relative to total measured mass are in contrast to results from
a surface site in Metro Manila (Stahl et al., 2020), where most of the same organic species
(adipate, succinate, maleate, oxalate, MSA) accounted for < 1% of total aerosol mass. Therefore,
the first hypothesis of this study holds true that the contributions of measured organic species
account for a greater portion of total measured mass in cloud water as compared to surface
particulate matter.
Gravimetry was used to measure total mass in the surface measurements whereas in cloud water,
total measured mass was more restrictive in terms of being based on measurable species, thus
qualifying our percentages as an upper bound. However, the measured ions in cloud water should
contribute relatively more to total measured mass in cloud water owing to their hygroscopic
nature and greater ease to become associated with cloud water as compared to more hydrophobic
species like black carbon that contribute significantly to total aerosol mass in the boundary layer
of Metro Manila. For example, black carbon accounted for 78.1% and 51.8% of the total mass


between 0.10 – 0.18 μm and 0.18 – 0.32 μm in Metro Manila surface aerosol particles (Cruz et
al., 2019), respectively, size ranges of which are highly relevant to droplet activation. Air masses
aloft in the CAMP$^2$Ex region, and especially those processed by clouds, are likely more aged and
oxidized compared to fresh organic emissions (e.g., automobiles, industry, burning) in the
surface layer over Metro Manila, which is the most populated urban area within the CAMP$^2$Ex
flight domain. Recent work has shown that cloud processing of isoprene oxidation products (a
key fraction of organic precursor vapors involved with organic aerosol generation) is the main
source of secondary organic aerosol (SOA) in the mid-troposphere (4 – 6 km) (Lamkaddam et
al., 2021). This motivates examining vertical TOC and organic species characteristics in more
detail, which is discussed next.

### 351   3.2 Vertical Profiles

The vertical profile of TOC levels was of interest as it relates to general vertical distribution of
organic matter in the troposphere. Measurements off the coast of Japan approximately two
decades ago during the ACE-Asia campaign revealed unexpectedly high organic aerosol levels in
the free troposphere due to presumed SOA formation (Heald et al., 2005). During that campaign,
organic aerosol concentrations in the boundary layer and free troposphere, and their relative
contribution to total non-refractory aerosol mass (organic, $SO_4^{2-}$, $NO_3^-$, $NH_4^+$), were amongst the
highest of various global regions examined (Heald et al., 2011). Therefore, it is of interest to
examine such types of vertical profiles farther south in the CAMP$^2$Ex region where data are
more scarce, with the unique aspect of this work being the focus on cloud water composition.
The highest TOC levels were observed in the bottom two kilometers, with a general reduction
above that altitude (Fig. 3). The decrease of TOC concentration with respect to altitude could be
attributed to more dilution in larger droplet sizes; results of cloud microphysical data will be the
focus of forthcoming work. Four data points influenced by biomass burning were singled out in
red markers (Fig. 3a) owing to having systematically higher TOC levels than other points. Those
points will be discussed in more detail in Sect. 4, and it is noteworthy that clouds were impacted
by biomass burning across a wide range of altitudes up to almost 7 km.
Focusing on the non-biomass burning (non-BB) data, there was considerable variation in the
bottom 2 km in TOC, with levels as low as 0.144 ppm C and as high as 3.362 ppm C.
Interestingly, cloud water collected above 5 km tended to still show enhanced TOC levels,
reaching up to 1.530 ppm C (6.1 km) among the non-BB points. The composition contributing to
TOC was similar with altitude in non-BB and biomass burning (BB) conditions, with ~75% of
TOC mass unaccounted for by the measured species, and MCAs dominating the measured
organic mass (Fig. 3b). The exception to that was the high-altitude BB point where 95.6% of
TOC was unassigned. Fig. 3c-d show that there was some qualitative agreement in the vertical
profile of AMS organic and m/z 44 for data collected immediately adjacent to the cloud water
samples in cloud-free air; more specifically, the highest levels of AMS organic, m/z 44, and TOC
were in the bottom 2 km. However, some interesting differences exist as they related to specific
air mass types as will be discussed in Sect. 4. Some differences could be rooted in how AMS





data represent submicrometer particles whereas cloud water data encompass a wider range of
particle sizes that activated into cloud droplets (including supermicrometer dust and sea salt
particles) and also gases partitioning to cloud water.
Vertical profiles of ratios representative of the relative amount of oxidized organics are shown in
Fig. 4. The MO:TOC ratio was quite variable with altitude ranging from 0.16 to 0.32 based on
the locally averaged curve shown; individual sample values ranged from 0.01 to 0.92. Vertically-
resolved ratio values for $f_{44}$ in cloud-free air and in cloud (downstream CVI) ranged on average
between 0 to 0.35 and 0.13 to 0.35, respectively. While mass concentrations decreased with
altitude (Fig. 3), ratios relevant to the degree of organic aerosol oxidation and make-up of the
organic component of cloud water did not exhibit a clear change with altitude.

## 4. Case Studies

Four subsets of samples are examined here to probe how the organic nature of cloud water varies
for distinct air masses. Sources of the air masses are visually shown in Fig. 5 based on 5-day
HYSPLIT back-trajectories: (i) "North" (RF11, n = 20) collected off the northern coast of Luzon
with influence from East Asia, the Korean Peninsula, and Japan; (ii) "East" (RF13, n = 11)
collected off the eastern coast of Luzon with back-trajectories traced to southern China with
subsequent passage across Luzon before arriving to the area of sample collection; (iii) "Biomass
Burning" (RF09, n = 4) collected to the southwest of Luzon above the Sulu Sea with influence
from biomass burning plumes from Borneo and Sumatra primarily consisting of peat as the fuel
type (Field and Shen, 2008; Levine, 1999; Page et al., 2002; Stockwell et al., 2016; Xian et al.,
2013); and (iv) "Clark" (RF04, RF06, RF07, RF09, RF10, and RF11, n = 25) collected around
the operational area over Luzon, approximately ~90 km northwest of Metro Manila, with back-
trajectories extending to the west and southwest of Luzon.
Biomass burning samples were identified based on the following criteria: flight scientist notes,
elevated surface smoke concentrations and aerosol optical depth (AOD) from the NAAPS model,
and the remarkable enhancement in chemical concentrations in cloud water. TOC, $K^+$, $SO_4^{2-}$, and
$NH_4^+$ in particular were enhanced in these samples with levels exceeding 4 ppm C, 0.13 ppm, 2.3
ppm, and 2.5 ppm, respectively.
Vertical profile results shown previously (Figs. 3-4) show markers corresponding to these four
case studies. With the exception of one BB sample collected at 6.5 km, samples in the four cases
were obtained below 3.3 km.

### 4.1 North

This category of samples was unique in that the mean MO (527.48 ± 301.59 ppb) and TOC (636
± 230 ppb C) concentrations were the lowest of all four cases (Table 2). The largest three organic
contributors to TOC were acetate (177.82 ± 72.96 ppb C; 11.5 ± 4.0%), oxalate (148.67 ± 81.47
ppb C; 6.0 ± 1.3%), and formate (83.16 ± 79.65 ppb C; 3.0 ± 2.2%). Maleate and DMA were not
detected for this case and 73.3% of the TOC went unaccounted for. Samples in this category





were collected between 1.2 and 2.9 km, without any pronounced organic chemical trends with
altitude (Figs. 3-4).
This case exhibited a few distinct features worth noting. First, it had the highest sea salt presence
based on the highest case-wide levels of $Na^+$ (3238 ± 2861 ppb), $Cl^-$ (5277 ± 4333 ppb), $Mg^{2+}$
(347 ± 328 ppb), and $Br^-$ (16 ± 8 ppb), the latter of which is a trace component of sea salt
(Seinfeld and Pandis, 2016). MSA originates partly from marine emissions of DMS, but its
concentration was among the lowest of all species for all four cases with a mass contribution to
total TOC (based on C mass) of only 0.17 ± 0.05% in the North category (Table 3). In their
analysis of aerosol data in the surface layer of Metro Manila, Stahl et al. (2020) showed lower
overall organic acid aerosol concentrations in the northeast monsoon season where northeasterly
air masses originated predominantly from East Asia; Stahl et al. (2020) also showed those air
masses were characterized by an enhancement in organic acid levels in the supermicrometer size
range owing to adsorption to coarse particle types such as sea salt and dust, but with a preference
for dust (Mochida et al., 2003; Rinaldi et al., 2011; Sullivan and Prather, 2007; Turekian et al.,
2003). As there was no direct evidence of dust in this case as the $Ca^{2+}$:$Na^+$ ratio was on average
(0.04) nearly the same as sea salt (0.038) (Seinfeld and Pandis, 2016), organic acids could have
interacted with sea salt. There were strong correlations between sea salt constituents, TOC, and
almost all detected organics (Table S1).
The second notable feature of this case was limited air mass aging characteristics based on
speciated ratios. The acetate:formate ratio is often used to indicate the relative influence of fresh
emissions (higher ratios) as compared to secondary production (lower ratios) (Talbot et al., 1988;
Wang et al., 2007). In at least one study, fresh emissions were linked to cloud water ratios above
1.5 and aged samples having values below 1 (Coggon et al., 2014). The mean acetate:formate
ratio for this air mass category was 4.21 ± 3.26, which was the highest of all four categories in
Table 2, suggestive of fresh emissions and low aging. This was consistent with the $Cl^-$:$Na^+$ ratio
(1.70 ± 0.13) being the close to sea water (1.81); our use of this ratio in the study assumes these
species originate primarily from sea salt. Lower $Cl^-$:$Na^+$ values in the study region coincide with
sea salt reactions with acids such as sulfuric, nitric, and organic acids (AzadiAghdam et al.,
2019). This was one of the two cases that had adipate present, with this category exhibiting the
highest mean concentration (5.15 ± 6.27 ppb). This suggests there was influence from cyclic
organics possibly originating from combustion sources, among others, during the transport to the
sample region. Adipate exhibited negative correlations with almost all other organic species in
this case (r: -0.48 – -0.72), suggestive of limited aging to form shorter chain carboxylic acids via
photochemical reactions (Table S1). With the exception of adipate, interrelationships between
the other organics detected in this case exhibited positive and significant correlations with one
another suggestive of common precursors and/or production mechanisms. Therefore, the results
of the North case point to influences from marine emissions and limited aging signatures based
on speciated ratios.

**4.2 East**





The dominant organic contributors to TOC (1051 ± 331 ppb C) in the East case were the same as
the North case with the difference being the order after acetate: acetate (359.04 ± 40.71 ppb; 14.9
± 3.1%), formate (258.18 ± 122.19 ppb; 7.2 ± 3.8%), and oxalate (153.63 ± 81.06 ppb; 3.8 ±
1.2%). The percentage of TOC unaccounted for by the speciated measurements (69.4%) was the
lowest out of all of the cases. This case resembled the North one in that there was marine
influence, but with differences being more pronounced dust influence and greater evidence of
aging based on chemical ratios. Marine signatures come from the second highest levels of $Na^+$,
$Cl^-$, and $Mg^{2+}$ after North, with high correlations between these species (Table S2).
Unlike the previous case, the $Ca^{2+}$:$Na^+$ ratio (0.10) was elevated from that of typical sea salt
(0.038). Wang et al. (2018) showed that East Asian dust can get lofted up during dust storms,
which could contribute to the transport to the Philippines. Previous studies have shown that
organic acids adsorb more readily to dust as compared to sea salt due to dust's more alkaline
nature (Stahl et al., 2020; Sullivan and Prather, 2007). While $Ca^{2+}$ was correlated to six of the 11
organic species for this case (r: 0.70 – 0.96; Table S2), the magnitude of the correlations was
very similar to those between either $Na^+$ or $Cl^-$ and the speciated organics. TOC also exhibited
similar correlations with $Na^+$, $Cl^-$, and $Ca^{2+}$ (r: 0.83 – 0.87). Therefore, it is too difficult with the
given data to assert whether (if at all) the organic acids had a preference towards either salt or
dust aerosol particles; of note though is that oxalate exhibited the strongest correlation with
either $Na^+$, $Cl^-$, and $Ca^{2+}$ (r: 0.96 – 0.99) among all species and also TOC. Additionally, Park et
al. (2004) showed enhanced $Ca^{2+}$ and $NO_3^-$ in the coarse mode owing to continental Asian dust.
In the East case, speciated organics were fairly well correlated to $NO_3^-$ (r: 0.68 – 0.99), which
has been associated with adsorption onto coarse aerosol types like dust and sea salt (e.g.,
Maudlin et al., 2015; Stahl et al., 2020). Nitrate was especially well correlated with $Na^+$, $Cl^-$, and
$Ca^{2+}$ (r: 0.98 – 1.00), which exceeded correlations of other common inorganic ions such as $SO_4^{2-}$
and $NH_4^+$.
The vertical profiles show clearly the systematically higher TOC levels relative to the North case
across roughly the same altitude range (1.3 – 3.3 km), but in contrast the AMS organic and m/z
44 values (although sparse) were more comparable, which again can simply be due to the
differences in what is being measured with AMS not accounting for the supermicrometer
particles types (i.e., dust and sea salt) that likely were more influential in the cloud water in the
East case.
Evidence of greater aging as compared to the North case comes from a few ratios of interest. The
$Cl^-$:$Na^+$ ratio for this case (1.40 ± 0.06) was lower than the North case, suggestive of more sea
salt reactivity aided by presumed aging. Furthermore, the acetate:formate ratio (1.93 ± 1.51) was
less than half the value from the North case. More broadly, the overall contribution of MCAs and
DCAs to TOC were very similar between the North and East cases and also the next two cases:
MCA:TOC = 16.03% – 23.66%, and DCA:TOC = 3.70% – 8.75% (Table 3). In contrast to the
North case, this category of samples had weaker interrelationships between organic species
presumed to be due to the mixture of sources impacting this case including dust, marine
particles, and likely other anthropogenic and biogenic sources over land.


**4.3 Biomass Burning**

The BB category samples exhibited the highest levels of TOC ($8342 \pm 3730$ ppb C) and almost every organic with the dominant contributors to TOC being formate ($2177.50 \pm 1588.91$ ppb; $7.0 \pm 4.5\%$), acetate ($1845.21 \pm 1667.91$ ppb; $8.4 \pm 5.6\%$), and succinate ($557.00 \pm 575.65$ ppb; $2.4 \pm 1.7\%$). As acetate and formate were so abundant, the relative enhancement of MCA mass was much larger than DCA mass as compared to the three other cases examined (Table 2). While the correlation matrix for this case was quite sparse in terms of significant values owing partly to such few points ($n = 4$), TOC and $K^+$ were highly correlated (r: 0.99), which demonstrates the strong linkage between TOC and biomass burning emissions (Table S3) as also shown by others (Cook et al., 2017). For context, Desyaterik et al. (2013) reported cloud water TOC levels of 100.6 ppm C in a biomass burning airmass at Mt. Tai in eastern China that was eight times higher than typical values in the absence of agricultural burning. Cook et al. (2017) observed significant higher cloud water TOC levels during wildfire periods at Whiteface Mountain, New York (16.6 ppm C) than biogenic (2.16 ppm C) or urban (2.11 ppm C) periods.

In our BB samples, mean values of succinate ($557.00 \pm 575.65$ ppb), glutarate ($150.39 \pm 82.20$ ppb), and pyruvate ($125.93 \pm 126.12$ ppb) were significantly elevated above the other cases. Stahl et al. (2020) recently showed that succinate, oxalate, and MSA were especially enhanced in aerosol samples collected in the study region during BB periods in the 2018 southwest monsoon season. Study-wide peak levels of succinate (1372.00 ppb), oxalate (1135.00 ppb), and MSA (24.79 ppb) were found in this case reinforcing those findings (Stahl et al., 2020). Unlike the previous two cases, maleate was detected in BB samples ($5.58 \pm 6.46$ ppb). Although maleate is associated with combustion sources (Kawamura and Kaplan, 1987; Rogge et al., 1993), such as from extensive ship traffic around the sampling area, other studies have shown enhancements of maleate in BB air masses (i.e., Mardi et al., 2019; Tsai et al., 2013). The percentage of mass contributing to TOC that was unaccounted for was 78.7%, with the highest sample at 6.5 km having 95.6% undetected, which was surprisingly large based on the prevalence of organic acids in biomass burning emissions (Reid et al., 1998). Therefore, the second hypothesis posed in this study is partly true in that the BB case exhibited much higher TOC values; however, these samples did not exhibit a greater contribution by organic acids to TOC since the North and East cases actually had a greater contribution from such species. This motivates more attention to organic chemical speciation in clouds impacted by biomass burning emissions as such a large portion of the TOC mass went unaccounted for in this study.

While absolute concentrations of most organics were greatly enhanced, the relative contributions of individual organics within the MCA and DCA subsets of species also varied. Most notably in the MCA category, formate was greatly enhanced with a mass contribution to total MCA mass being 46.40% versus 16.54% – 29.09% for other cases. In the DCA population of species, glutarate (17.15% versus 0.65% – 4.02%) and succinate (41.95% versus 20.82% – 38.52%) accounted for a higher mass fraction than other cases.

The $Cl^-$:$Na^+$ ratio was $1.30 \pm 0.06$ and suggestive of $Cl^-$ depletion, which has been observed in other regions with biomass burning and linked to high levels of inorganic and organic acids (Braun et al., 2017, and references therein). This is supported by how the values of MO, $SO_4^{2-}$,





and $NO^{3-}$ were the highest in this case (Table 2). The acetate:formate ratio was $0.69 \pm 0.30$, but it
is unclear as to how effective this and other ratios are as aging indicators when biomass burning
is present and especially as fuel type varies between regions.

**4.4 Clark**

Samples in this category were collected during ascents after takeoff and descents during
approaches to the airfield, which allowed for sample collection closer to the surface than the
other categories (altitude range: 0.2 – 2.9 km). Clark International Airport is located within the
Clark Freeport Zone, which is part of both the Pampanga and Tarlac provinces and consists of
five cities and municipalities: Angeles City, Mabalacat City, Porac, Capas, and Bamban. This
gives the Clark area a population of approximately 996,000 with a population density of ~3100
$km^{-2}$, which is low in comparison to the most populated city in the Philippines, Quezon City in
Metro Manila, with 2.94 million people and a population density of ~17000 $km^{-2}$ (PSA, 2016). In
addition to Metro Manila just to the southeast (~90 km), Clark lies between Mt. Pinatubo to the
west and Mt. Arayat to the east, which are active and potentially active volcanoes, respectively.
The average TOC for this case ($1181 \pm 920$ ppb C) was most similar to the East case and
exhibited the most variability relative to the mean TOC value of all four cases, which we
attribute to numerous sources impacting these samples including local and regional emissions,
time of day variability, local spatial variability, and number of flights. This case exhibited the
highest percentage of TOC mass unaccounted for by speciated organics (79.5%) with the three
largest measured contributors consisting of acetate ($296.65 \pm 325.80$ ppb; $9.6 \pm 9.5$%), formate
($266.05 \pm 316.80$ ppb; $4.8 \pm 3.3$%), and oxalate ($88.33 \pm 103.88$ ppb; $1.7 \pm 1.0$%). A few notable
features are mentioned specific to this case. This was the only case that had DMA present ($6.45$
$\pm 15.89$ ppb) albeit with a low mass contribution to total TOC ($0.43 \pm 1.17$%). This case
exhibited the highest mass fractions of maleate ($3.20 \pm 5.93$%) and adipate ($16.05 \pm 21.48$%)
relative to DCA mass, suggestive of greater anthropogenic emission influence and processed
aromatic compounds. DMA was only correlated with maleate (r: 0.67) among the organic
species suggestive of a similar source (Table S4). Stahl et al. (2020) showed increased aerosol
concentrations of freshly emitted organics (i.e., phthalate, maleate) owing to the vast sources of
combustion engines to the southeast of the Clark area. Clark is situated near a major highway
that could also contribute to the high combustion sources, though commercial aircraft emissions
could also have a significant role.
Because succinate peaked in concentration for this case (498.50 ppb) and back-trajectories
originated from Borneo and Sumatra, there may have been some influence from biomass burning
(Fig. 5). The $K^+$:$Na^+$ ratio was elevated (0.25) above that of sea salt (0.036) (Seinfeld and Pandis,
2016), and even higher than the Biomass Burning case (0.15), suggestive of local and/or regional
biomass burning influence. This case exhibited the highest mean $Ca^{2+}$:$Na^+$ ratio (0.99) that was
well above the sea salt value (0.038), which we presume could be linked largely to resuspended
and/or transported dust. Cruz et al. (2019) showed for Metro Manila that resuspended dust,
especially linked to vehicular traffic, is an important source of dust in the study region. Stahl et



al. (2020) showed that adipate is most influenced by crustal sources in the study region and was
unique among the studied organics in this work in that it exhibited a prominent peak in the
supermicrometer range based on surface aerosol measurements in Metro Manila. Consistent with
that work, $Ca^{2+}$ was only correlated with adipate in the Clark samples (r: 0.71) among the studied
organics (Table S4), adding support for how organic acids like adipate can partition to dust with
the novelty here being that the signature was observed in cloud water.

**5.  Conclusion**
This work analyzed 159 cloud water samples collected over a 2-month period as part of the
CAMP$^2$Ex airborne campaign around the Philippines. TOC and a total of eleven organic
compounds comprised of four MCAs (glycolate, acetate, formate, and pyruvate), five DCAs
(glutarate, adipate, succinate, maleate, and oxalate), MSA, and DMA were measured. The
measured organics were then compared to TOC to determine the percentage of organic species
measured compared to the total organic composition. Notable results are summarized below
including responses to the two hypotheses proposed at the end of Sect. 1.
• TOC levels ranged widely between 0.018 – 13.660 ppm C between 0.2 – 6.8 km, with a
mean value of 0.902 ppm C. The contribution (in C mass) of the 11 measured species to
total TOC was on average 30%. Using a conversion factor of 1.8 for organic matter
relative to organic carbon, the mean amount of total organic matter (TOM) accounted for
by our measured 11 species was 46.4%. Furthermore, the mean contribution of TOM and
speciated organics to total mass (inorganics + organics) was 30.7% (maximum = 95.1%)
and 10.3% (maximum = 57.6%), respectively. The mean ratio of inorganic to TOM was
5.8. The study's first hypothesis holds true that the measured organic species account for
a higher mass fraction relative to total mass as compared to surface layer aerosol
measurements over Luzon (< 1%), (Stahl et al., 2020). This is likely owing to more
processed air masses aloft and the reduced influence of black carbon that is so abundant
in areas like Metro Manila (Cruz et al., 2019; Hilario et al., 2020a).
• In terms of the chemical profile of the speciated organics, the order in decreasing
contribution of C mass relative to TOC was as follows: acetate (14.7 ± 20.5%), formate
(5.4 ± 9.3%), oxalate (2.8 ± 4.3%), DMA (1.7 ± 6.3%), succinate (1.6 ± 2.4%), pyruvate
(1.3 ± 4.5%), glycolate (1.3 ± 3.7%), adipate (1.0 ± 3.6%), MSA (0.1 ± 0.1%), glutarate
(0.1 ± 0.2%), maleate (< 0.1 ± 0.1%). Approximately 70.0% of TOC went unaccounted
for pointing to the complexity and difficulty of organic speciation in the study region,
with this value fairly similar to other regions too (Benedict et al., 2012; Boris et al., 2016;
Boris et al., 2018; Herckes et al., 2002; Raja et al., 2008). Monocarboxylic acids
dominated the speciated organic mass (~75%) and were about four times more abundant
than dicarboxylic acids, suggestive of higher abundance of gaseous species and
precursors.
• Vertical profiles of TOC revealed higher levels in the bottom 2 km with a reduction
above that. Samples impacted by biomass burning emissions were significantly enhanced
in TOC and most speciated organic levels, ranging in altitude from as low as 1.3 km to as



high as 6.5 km. While vertical profiles of AMS organic and m/z 44 qualitatively
resembled that of TOC with reductions above 2 km, the vertical behavior of chemical
ratios relevant to the composition of the cloud (ratio of C mass from measured organics
to TOC) and aerosol organics ($f_{44}$) did not reveal any clear trend. For both non-BB and
BB samples, monocarboxylic acids uniformly dominated C mass with ~75% of TOC
mass unaccounted for across the range of altitudes studied.
• The second hypothesis in this study proved to be partly true as clouds impacted by
biomass burning exhibited markedly higher values of TOC (4.974 – 13.660 ppm C) and
masses of most all other species detected as compared to the other three categories of
samples in Sect. 4 (North, East, Clark). However, the part of the hypothesis about
speciated organic acids contributing more to BB samples did not hold true as total
measured organics accounted on average for 21.25% of the TOC, which was lower than
two of the other categories of samples (North [26.72%] and East [30.61%]). Interestingly,
the highest BB sample (6.5 km) had 95.6% of the C mass unaccounted for by speciated
organics. This motivates increased attention to organic speciation in clouds impacted by
biomass burning.
• Four categories of samples with different air mass history characteristics were compared
revealing a few notable features: (i) while speciated concentrations and TOC levels
varied considerably between the four cases, the contributions of MCAs and DCAs (based
on C mass) to TOC were remarkably similar with little variation (MCA:TOC = 16.03% –
23.66%, DCA:TOC and 3.70% – 8.75%); (ii) dust and sea salt tracer species were
strongly correlated to most all speciated organics for the North and East cases suggestive
of interactions between such species and coarse aerosol surfaces as supported by past
work (Stahl et al., 2020; Sullivan and Prather, 2007); (iii) for samples with limited aging
(North case) based on selected chemical ratio values, adipate was more abundant and
negatively correlated to smaller carboxylic acids; (iv) BB samples exhibited the highest
TOC concentrations (8342 ± 3730 ppb C) as well as significant elevations in individual
organics such as acetate, formate, succinate, glutarate, pyruvate, oxalate, and MSA; and
(v) the Clark case had a higher variability of TOC (1181 ± 920 ppb C) compared to the
North and East cases presumably owing to a greater mix of influential sources such as
fresh anthropogenic emissions (e.g., enhanced maleate), but also transport of biomass
burning plumes from Borneo and Sumatra (e.g., enhanced succinate), dust, as well as
spatial and temporal variances across different flights. Related to dust, $Ca^{2+}$ was only
correlated to adipate in the Clark samples, consistent with a recent study in Metro Manila
(Stahl et al., 2020) showing that adipate uniquely exhibits a prominent supermicrometer
peak among organic acids attributed to interactions with dust.

## Data availability

All data used can be found on the NASA data repository at
DOI:10.5067/Suborbital/CAMP2EX2018/DATA001.





**Author contributions**

EC, RAB, CS, ABM, and AS designed the experiment. All coauthors carried out various aspects of the data collection. EC, CS, and AS conducted analysis and interpretation of the data. CS and AS prepared the manuscript with contributions from the coauthors.

**Competing interests**

The authors declare that they have no conflict of interest.

**Acknowledgements**

The authors acknowledge support from NASA grant 80NSSC18K0148 in support of the NASA CAMP$^2$Ex project. R. A. Braun acknowledges support from the ARCS Foundation. M. Cruz acknowledges support from the Philippine Department of Science and Technology's ASTHRD Program. A. B. MacDonald acknowledges support from the Mexican National Council for Science and Technology (CONACYT).

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



**Table 1:** Mass concentration limits of detection (LOD), minimum, maximum, mean, one
standard deviation, and median values (ppb; left), in addition to mass fraction (%; right) for the
159 CAMP²Ex cloud water samples with TOC data; note that mass fraction values depend on the
C mass of each organic species shown. Total measured mass is defined as the sum of TOM, $Na^+$,
$NH_4^+$, $K^+$, $Mg^{2+}$, $Ca^{2+}$, $Cl^-$, $NO_2^-$, $Br^-$, $NO_3^-$, and $SO_4^{2-}$. MCA – monocarboxylic acids, DCA –
dicarboxylic acids, MSA – methanesulfonate, DMA – dimethylamine, MO – measured organics,
TOM – total organic matter, DL – detection limit.

| | LOD | Concentration (ppb) | | | | | Mass Fraction (%) | | | | |
|---|---|---|---|---|---|---|---|---|---|---|---|
| | | Min | Max | Median | Mean | Stdev | Min | Max | Median | Mean | Stdev |
| Glycolate | 98.76 | <DL | 224.8 | 10.7 | 13.5 | 20.3 | 0.0 | 35.0 | 0.6 | 1.3 | 3.7 |
| Acetate | 6.38 | <DL | 3926.0 | 159.4 | 251.4 | 409.9 | 0.0 | 100.0 | 10.5 | 14.7 | 20.5 |
| Formate | 19.77 | 2.1 | 3819.0 | 66.6 | 188.5 | 432.5 | 0.2 | 100.0 | 3.8 | 5.4 | 9.3 |
| Pyruvate | 5.45 | <DL | 296.9 | 5.4 | 24.4 | 41.3 | 0.0 | 56.1 | 0.5 | 1.3 | 4.5 |
| MCA | - | 13.4 | 8041.9 | 253.4 | 477.8 | 857.8 | 0.6 | 100.0 | 16.9 | 22.6 | 33.9 |
| Glutarate | 43.70 | <DL | 258.7 | <DL | 6.8 | 27.3 | 0.0 | 1.0 | 0.0 | 0.1 | 0.2 |
| Adipate | 39.21 | <DL | 71.5 | 3.0 | 5.3 | 8.3 | 0.0 | 43.7 | 0.4 | 1.0 | 3.6 |
| Succinate | 38.64 | <DL | 1372.0 | <DL | 55.2 | 137.7 | 0.0 | 9.3 | 0.0 | 1.6 | 2.4 |
| Maleate | 14.81 | <DL | 14.7 | <DL | 0.7 | 2.3 | 0.0 | 0.8 | 0.0 | 0.0 | 0.1 |
| Oxalate | 55.23 | <DL | 1135.0 | 38.6 | 95.6 | 148.2 | 0.0 | 43.9 | 1.7 | 2.8 | 4.3 |
| DCA | - | 1.5 | 2765.7 | 61.4 | 163.7 | 295.3 | 0.1 | 69.8 | 3.3 | 5.5 | 7.5 |
| MSA | 88.01 | <DL | 24.8 | 3.9 | 5.1 | 5.3 | 0.0 | 0.9 | 0.1 | 0.1 | 0.1 |
| DMA | 56.97 | <DL | 183.8 | <DL | 11.2 | 32.4 | 0.0 | 45.3 | 0.0 | 1.7 | 6.3 |
| MO | - | 29.5 | 10815.3 | 334.3 | 657.7 | 1124.7 | 1.5 | 100.0 | 23.8 | 30.0 | 41.2 |
| TOC | 0.05 | 18 | 13660 | 546 | 902 | 1435 | ↑ *Relative to TOC (%)* ↑ | | | | |
| Inorg/TOM | - | 0.1 | 90.3 | 3.3 | 5.8 | 8.6 | Relative to total measured concentrations (%) | | | | |
| MO | - | - | - | - | - | - | 0.8 | 57.6 | 7.2 | 10.3 | 9.2 |
| TOM | - | 32 | 24588 | 983 | 1624 | 2584 | 1.1 | 95.1 | 23.2 | 30.7 | 24.5 |
| Inorganic | - | 26 | 117933 | 3894 | 8651 | 13645 | 4.9 | 98.9 | 76.8 | 69.3 | 24.5 |
| Na | 16.62 | <DL | 29280 | 609 | 1650 | 3192 | 0.0 | 26.6 | 9.5 | 10.0 | 7.8 |
| NH₄ | 176.80 | <DL | 8099 | 427 | 804 | 1010 | 0.0 | 68.1 | 7.2 | 11.2 | 13.2 |
| K | 142.35 | <DL | 1211 | 21 | 75 | 144 | 0.0 | 21.8 | 0.5 | 0.8 | 2.0 |
| Mg | 46.20 | <DL | 3701 | 58 | 182 | 379 | 0.0 | 4.0 | 1.0 | 1.1 | 0.9 |
| Ca | 74.81 | <DL | 1951 | 118 | 201 | 277 | 0.0 | 25.2 | 1.6 | 3.5 | 4.6 |
| Cl | 76.59 | <DL | 38200 | 908 | 2451 | 4438 | 0.0 | 42.7 | 15.3 | 16.0 | 11.7 |
| NO₂ | 46.24 | <DL | 16 | <DL | 2 | 3 | 0.0 | 0.4 | 0.0 | 0.0 | 0.1 |
| Br | 7.82 | <DL | 44 | 1 | 4 | 7 | 0.0 | 0.2 | 0.0 | 0.0 | 0.0 |
| NO₃ | 17.33 | <DL | 26560 | 572 | 1488 | 2925 | 0.0 | 43.4 | 10.4 | 12.6 | 8.2 |
| SO₄ | 414.73 | 2 | 15680 | 868 | 1795 | 2495 | 0.4 | 34.9 | 14.1 | 14.0 | 8.5 |



Atmospheric Chemistry and Physics Discussions — Open Access — EGU

**Table 2:** Speciated concentrations of organics (ppb) for each case study, where the first group of rows are monocarboxylic acids (MCA), the second group of rows are dicarboxylic acids (DCA), the third group of rows are other organics plus total measured organics (MO) and total organic carbon (TOC), inorganic ions, and the fifth group are select ratios. n = number of samples.

| | North (n = 20) | | | | | East (n = 11) | | | | | Biomass Burning (n = 4) | | | | | Clark (n = 25) | | | | |
|---|---|---|---|---|---|---|---|---|---|---|---|---|---|---|---|---|---|---|---|---|
| | Min | Max | Median | Mean | Stdev | Min | Max | Median | Mean | Stdev | Min | Max | Median | Mean | Stdev | Min | Max | Median | Mean | Stdev |
| Glycolate | 5.96 | 37.18 | 15.80 | 17.59 | 9.15 | 5.34 | 30.00 | 12.21 | 13.68 | 7.25 | <DL | 46.86 | 7.20 | 15.31 | 22.10 | <DL | 53.42 | 6.90 | 11.68 | 14.61 |
| Acetate | 1.94 | 288.80 | 184.85 | 177.82 | 72.96 | 301.70 | 423.50 | 358.60 | 359.04 | 40.71 | 47.85 | 3926.00 | 1703.50 | 1845.21 | 1667.91 | <DL | 1105.00 | 185.20 | 296.65 | 325.80 |
| Formate | 10.28 | 232.20 | 62.66 | 83.16 | 79.65 | 61.02 | 492.80 | 248.30 | 258.11 | 122.19 | 151.00 | 3819.00 | 2370.00 | 2177.50 | 1588.91 | 2.42 | 1041.00 | 152.00 | 266.05 | 316.80 |
| Pyruvate | 2.14 | 126.50 | 35.91 | 42.98 | 38.98 | 7.50 | 78.24 | 24.65 | 32.25 | 21.01 | <DL | 296.90 | 103.41 | 125.93 | 126.12 | 1.07 | 161.80 | 16.08 | 30.31 | 35.84 |
| MCA | 25.32 | 632.19 | 299.90 | 321.55 | 183.61 | 431.25 | 923.72 | 673.22 | 663.15 | 142.72 | 245.71 | 8041.90 | 4184.11 | 4163.96 | 3335.76 | 31.93 | 2066.20 | 369.18 | 604.69 | 641.90 |
| Glutarate | <DL | 10.18 | <DL | 1.53 | 2.76 | <DL | 10.86 | 4.07 | 5.12 | 3.67 | 62.46 | 258.70 | 140.20 | 150.39 | 82.20 | <DL | 62.46 | 1.36 | 9.42 | 16.85 |
| Adipate | <DL | 17.44 | <DL | 5.15 | 6.27 | <DL | <DL | <DL | <DL | <DL | <DL | <DL | <DL | <DL | <DL | <DL | 37.43 | <DL | 3.78 | 7.89 |
| Succinate | <DL | 136.20 | 28.09 | 42.30 | 47.84 | 15.45 | 176.90 | 63.90 | 74.11 | 57.27 | 24.58 | 1372.00 | 415.70 | 557.00 | 575.65 | <DL | 498.50 | 18.96 | 67.74 | 123.72 |
| Maleate | <DL | <DL | <DL | <DL | <DL | <DL | <DL | <DL | <DL | <DL | <DL | 11.61 | 5.36 | 5.58 | 6.46 | <DL | 14.73 | <DL | 2.71 | 4.17 |
| Oxalate | 37.53 | 330.40 | 124.75 | 148.67 | 81.47 | 52.51 | 311.20 | 123.30 | 153.63 | 81.06 | 303.80 | 1135.00 | 520.05 | 619.73 | 360.12 | 5.55 | 448.90 | 43.63 | 88.33 | 103.88 |
| DCA | 67.84 | 467.09 | 149.27 | 197.64 | 125.25 | 80.60 | 493.36 | 194.12 | 232.86 | 136.37 | 735.78 | 2765.70 | 914.65 | 1332.69 | 968.88 | 7.67 | 1009.86 | 72.10 | 171.99 | 238.04 |
| MSA | 1.55 | 14.72 | 7.75 | 8.29 | 3.16 | 3.10 | 17.82 | 10.07 | 10.57 | 4.40 | <DL | 24.79 | 3.87 | 8.13 | 11.69 | <DL | 10.85 | 3.87 | 4.18 | 3.62 |
| DMA | <DL | <DL | <DL | <DL | <DL | <DL | <DL | <DL | <DL | <DL | <DL | <DL | <DL | <DL | <DL | <DL | 61.25 | <DL | 6.45 | 15.89 |
| MO | 99.36 | 1088.83 | 483.78 | 527.48 | 301.59 | 567.01 | 1364.27 | 855.59 | 906.58 | 269.42 | 1016.58 | 10815.35 | 5093.60 | 5504.78 | 4186.65 | 54.05 | 3053.76 | 433.14 | 787.32 | 837.34 |
| TOC | 364 | 1085 | 555 | 636 | 230 | 663 | 1570 | 985 | 1051 | 331 | 4974 | 13660 | 7366 | 8342 | 3730 | 220 | 3362 | 849 | 1181 | 920 |
| Na | 693 | 11870 | 2273 | 3238 | 2861 | 618 | 6546 | 1970 | 2569 | 1738 | 833 | 4425 | 2160 | 2394 | 1624 | 12 | 5870 | 625 | 1105 | 1403 |
| NH4 | 181 | 1955 | 644 | 847 | 515 | 513 | 2379 | 1307 | 1432 | 587 | 2517 | 8099 | 3685 | 4496 | 2483 | 45 | 2880 | 947 | 1009 | 815 |
| K | 14 | 405 | 50 | 89 | 100 | 18 | 493 | 66 | 123 | 136 | 132 | 724 | 272 | 350 | 258 | 2 | 264 | 32 | 63 | 76 |
| Mg | 41 | 1338 | 236 | 347 | 328 | 62 | 668 | 209 | 273 | 183 | 84 | 501 | 242 | 267 | 191 | <DL | 631 | 65 | 117 | 153 |
| Ca | <DL | 765 | 97 | 174 | 219 | 49 | 778 | 167 | 269 | 219 | 106 | 534 | 236 | 278 | 210 | 39 | 904 | 177 | 230 | 184 |
| Cl | 1445 | 18520 | 3772 | 5277 | 4333 | 900 | 8357 | 2760 | 3510 | 2196 | 1126 | 5989 | 2553 | 3055 | 2124 | 45 | 9083 | 991 | 1716 | 2107 |
| NO2 | <DL | <DL | <DL | <DL | <DL | <DL | <DL | <DL | <DL | <DL | <DL | 6 | 1 | 2 | 3 | <DL | 16 | <DL | 3 | 5 |
| Br | 3 | 36 | 13 | 16 | 8 | 2 | 7 | 4 | 4 | 1 | 1 | 6 | 2 | 3 | 2 | <DL | 13 | 1 | 3 | 3 |
| NO3 | 477 | 5265 | 1197 | 1810 | 1506 | 1084 | 8724 | 2902 | 3772 | 2296 | 1880 | 7045 | 3344 | 3903 | 2277 | 65 | 2759 | 691 | 931 | 736 |
| SO4 | 1305 | 12120 | 3281 | 4503 | 2865 | 1212 | 5296 | 2819 | 3223 | 1385 | 2313 | 9993 | 4177 | 5165 | 3343 | 23 | 4406 | 1157 | 1416 | 1157 |
| Ace/For | 0.19 | 9.66 | 2.65 | 4.21 | 3.26 | 0.75 | 5.67 | 1.52 | 1.93 | 1.51 | 0.32 | 1.03 | 0.70 | 0.69 | 0.30 | 0 | 3.86 | 0.98 | 1.12 | 0.84 |
| Cl/Na | 1.52 | 2.08 | 1.69 | 1.70 | 0.13 | 1.28 | 1.51 | 1.40 | 1.40 | 0.06 | 1.07 | 1.43 | 1.35 | 1.30 | 0.16 | 1.38 | 3.70 | 1.69 | 1.84 | 0.56 |
| Ca/Na | 0 | 0.08 | 0.04 | 0.04 | 0.02 | 0.05 | 0.14 | 0.10 | 0.10 | 0.03 | 0.08 | 0.13 | 0.12 | 0.11 | 0.02 | 0.05 | 6.17 | 0.32 | 0.99 | 1.46 |
| K/Na | 0.02 | 0.03 | 0.02 | 0.02 | 0.01 | 0.03 | 0.08 | 0.04 | 0.04 | 0.01 | 0.10 | 0.18 | 0.16 | 0.15 | 0.04 | 0.01 | 3.93 | 0.05 | 0.25 | 0.78 |
| MO/TOC | 0.07 | 0.37 | 0.29 | 0.27 | 0.08 | 0.18 | 0.42 | 0.29 | 0.31 | 0.07 | 0.04 | 0.28 | 0.26 | 0.21 | 0.11 | 0.03 | 0.57 | 0.19 | 0.20 | 0.13 |





**Table 3:** Average organic composition for each case study where the first, second, and third
group of rows show percentage contribution (%) of individual components to monocarboxylic
acids (MCA), dicarboxylic acids (DCA), and total organic carbon (TOC), respectively.

| Group | Species (%) | North (n = 20) | | East (n = 11) | | BB (n = 4) | | Clark (n = 25) | |
|---|---|---|---|---|---|---|---|---|---|
| | | Mean | Stdev | Mean | Stdev | Mean | Stdev | Mean | Stdev |
| MCA | Glycolate | 7.20 | 9.20 | 1.84 | 0.81 | 5.09 | 9.87 | 17.65 | 29.05 |
| | Acetate | 64.03 | 17.74 | 64.20 | 10.85 | 45.86 | 14.07 | 46.35 | 23.98 |
| | Formate | 16.54 | 9.83 | 28.62 | 10.35 | 46.40 | 7.24 | 29.09 | 11.72 |
| | Pyruvate | 12.23 | 6.90 | 5.33 | 2.87 | 2.65 | 1.87 | 6.91 | 4.90 |
| DCA | Glutarate | 0.65 | 1.00 | 2.91 | 1.41 | 17.15 | 9.28 | 4.02 | 5.02 |
| | Adipate | 8.04 | 9.47 | 0 | 0 | 0 | 0 | 16.05 | 21.48 |
| | Succinate | 20.82 | 20.08 | 38.52 | 12.15 | 41.95 | 25.27 | 26.53 | 25.39 |
| | Maleate | 0 | 0 | 0 | 0 | 0.75 | 0.88 | 3.20 | 5.93 |
| | Oxalate | 70.49 | 12.29 | 58.57 | 11.52 | 40.16 | 16.50 | 50.20 | 17.42 |
| TOC | MSA | 0.17 | 0.05 | 0.13 | 0.04 | 0.01 | 0.02 | 0.06 | 0.07 |
| | DMA | 0 | 0 | 0 | 0 | 0 | 0 | 0.43 | 1.17 |
| | MCA | 17.79 | 6.17 | 23.66 | 5.99 | 16.03 | 10.13 | 16.28 | 11.91 |
| | DCA | 8.75 | 2.65 | 6.82 | 2.94 | 5.21 | 1.60 | 3.70 | 2.67 |
| | MO | 26.72 | 7.86 | 30.61 | 7.35 | 21.25 | 11.32 | 20.46 | 13.34 |
| | Undetected | 73.28 | 7.86 | 69.39 | 7.35 | 78.75 | 11.32 | 79.54 | 13.34 |




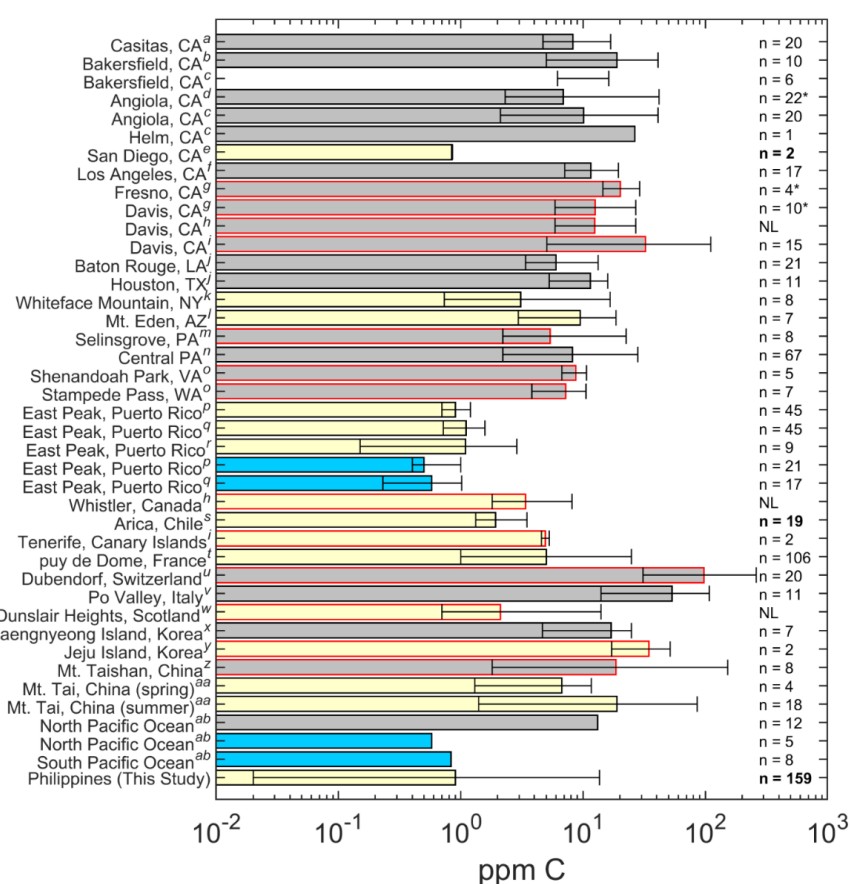


**Figure 1:** TOC (or DOC if TOC values were unavailable) concentrations reported for past
studies in relation to this work organized by continent. Bars represent the average values and the
error bars represent the minimum and maximum values. The absence of a solid bar means no
average was available. No error bars means there was no range given, and * indicates the median
value was reported rather than an average. Gray, yellow, and blue bars represent studies looking
at fog, clouds, and rain, respectively. Bars that are outlined in black are studies that used TOC
and bars outlined in red are studies that used DOC. The n values represent the number of samples
used in the study and NL means the number of samples were not listed. Bolded n values denote
airborne samples. (a - Boris et al. (2018), b - Collett Jr. et al. (1998), c - Herckes et al. (2002), d -
Herckes et al. (2007), e - Straub et al. (2007), f - Erel et al. (1993), g - Ehrenhauser et al. (2012),
h - Ervens et al. (2013), i - Zhang and Anastasio (2001), j - Raja et al. (2008), k - Cook et al.
(2017), l - Hutchings et al. (2008), m - Straub et al. (2012), n - Straub (2017), o - Anastasio et al.
(1994), p - Gioda et al. (2011), q - Gioda et al. (2008), r - Reyes-Rodríguez et al. (2009), s -
Benedict et al. (2012), t - Deguillaume et al. (2014), u - Capel et al. (1990), v - Gelencser et al.
(2000), w - Hadi et al. (1995), x - Boris et al. (2016), y - Decesari et al. (2005), z - Wang et al.
(2011), aa - Shen (2011), ab - Kim et al. (2020))



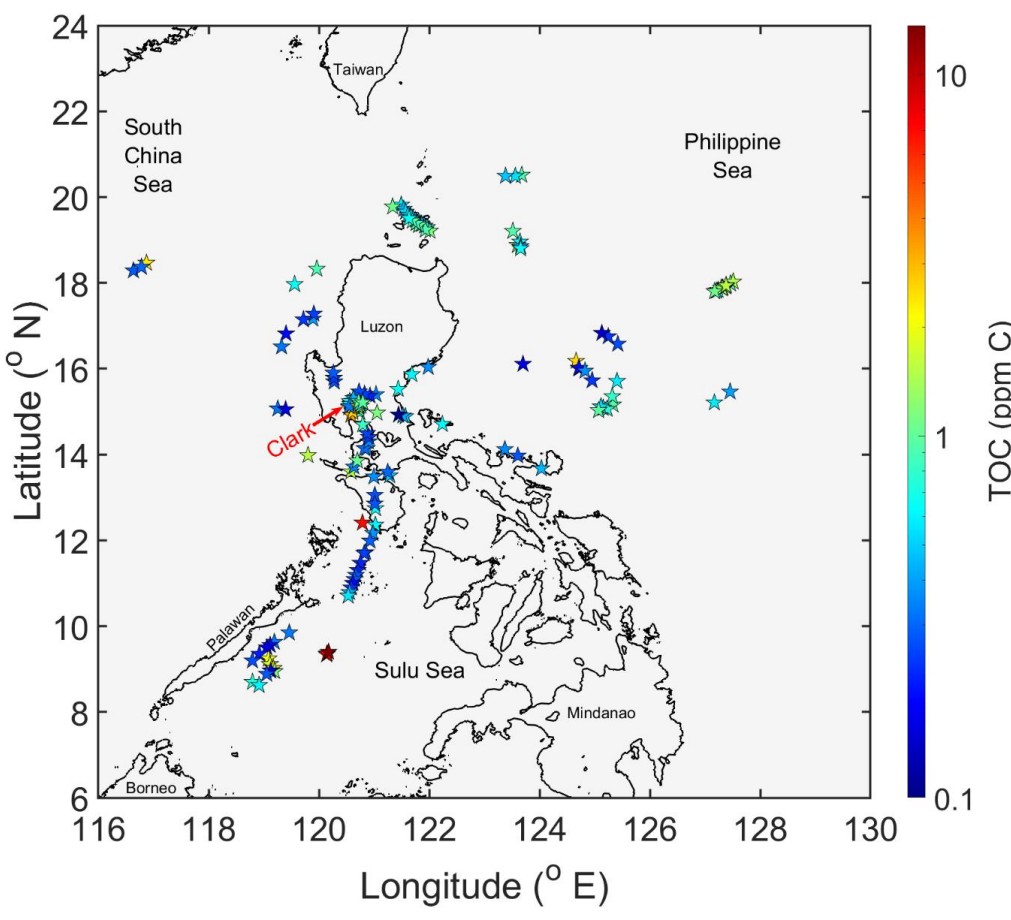

1100

**Figure 2:** Map of sample region where the stars represent the midpoint of the cloud water
samples where total organic carbon (TOC) was measured. Stars are colored by TOC on a
logarithmic scale.

1104



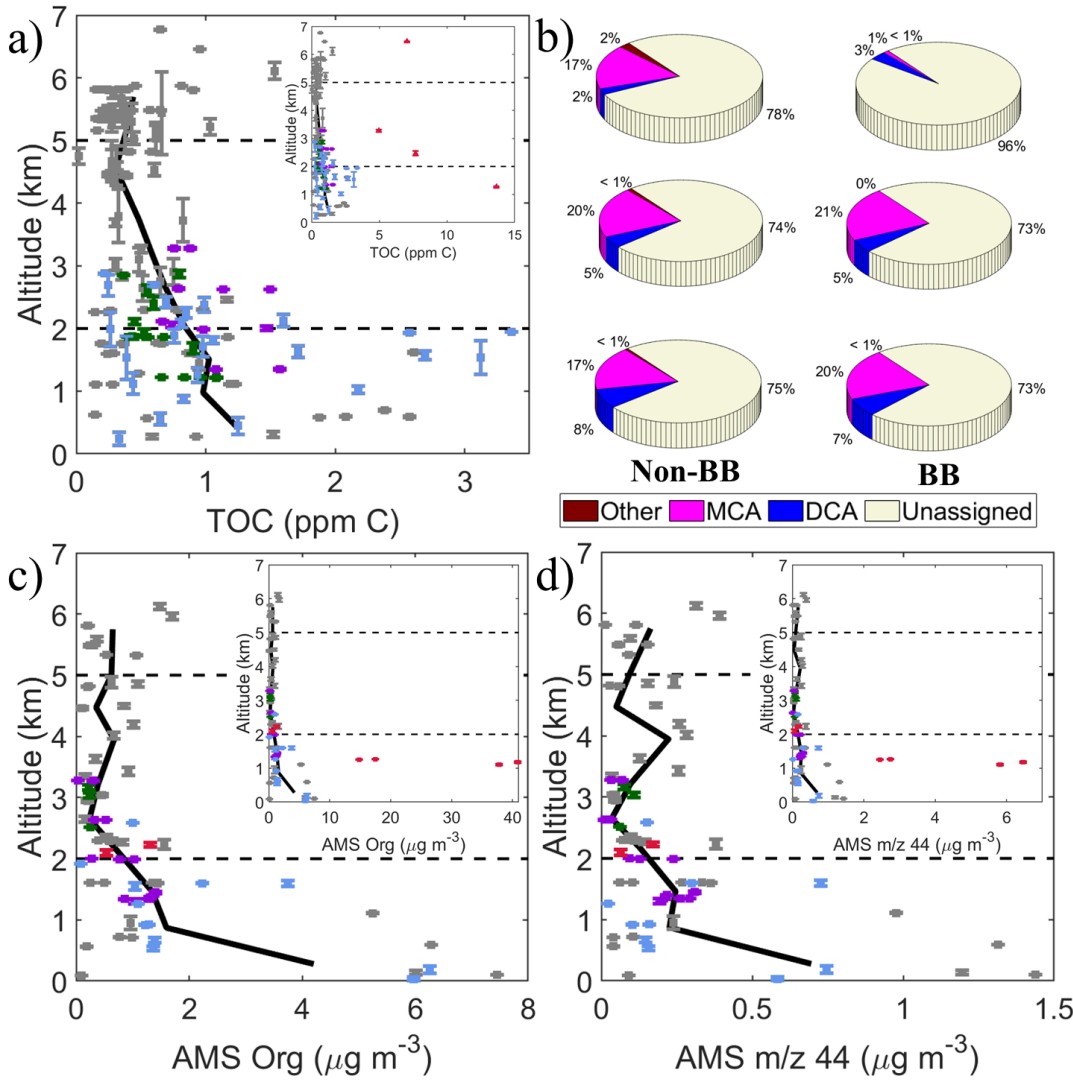

**Figure 3.** (a) Vertical profile of TOC concentrations (n = 159 samples) with the smaller inset including four samples with enhanced TOC owing to biomass burning (BB) influence. (b) Mass fractions of different subsets of species contributing to TOC at high (> 5 km), mid (2 − 5 km), and low (< 2 km) altitude with the beige area representing undetected species. Vertical profile of AMS (c) organic and (d) m/z 44 corresponding to spatially and temporally adjacent cloud-free periods of the collected cloud water samples. Colors in panels a/b/d represent the case study points in Sect. 4: North (green), East (purple), Biomass Burning (red), Clark (blue), non-case points (gray). The solid black lines in panels a/b/d represent locally-weighted average values. The error bars represent one standard deviation of the altitude variance.

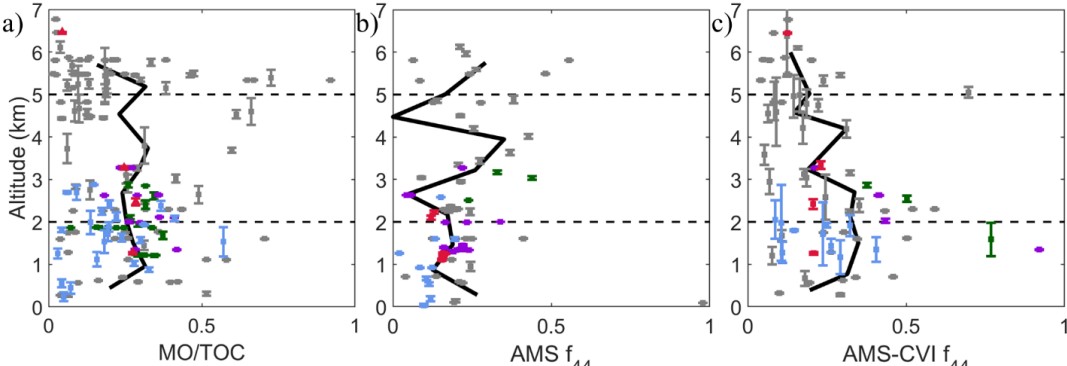

1115

**Figure 4.** Vertical profile of (a) ratio of C mass from measured organics (MO) to TOC for cloud
water samples, (b) AMS $f_{44}$ in cloud-free air, and (c) AMS-CVI $f_{44}$ in cloudy air. AMS data in (b)
corresponds to cloud-free periods that were spatially and temporally adjacent to the collected
cloud water samples, while those in (c) are within the period of cloud water collection times in
cloud. Colors in panels a/b/d represent the same case study points as Figure 3: North (green),
East (purple), Biomass Burning (red), Clark (blue), non-case points (gray). The black lines in
panels a/b/d represent locally-weighted average values.

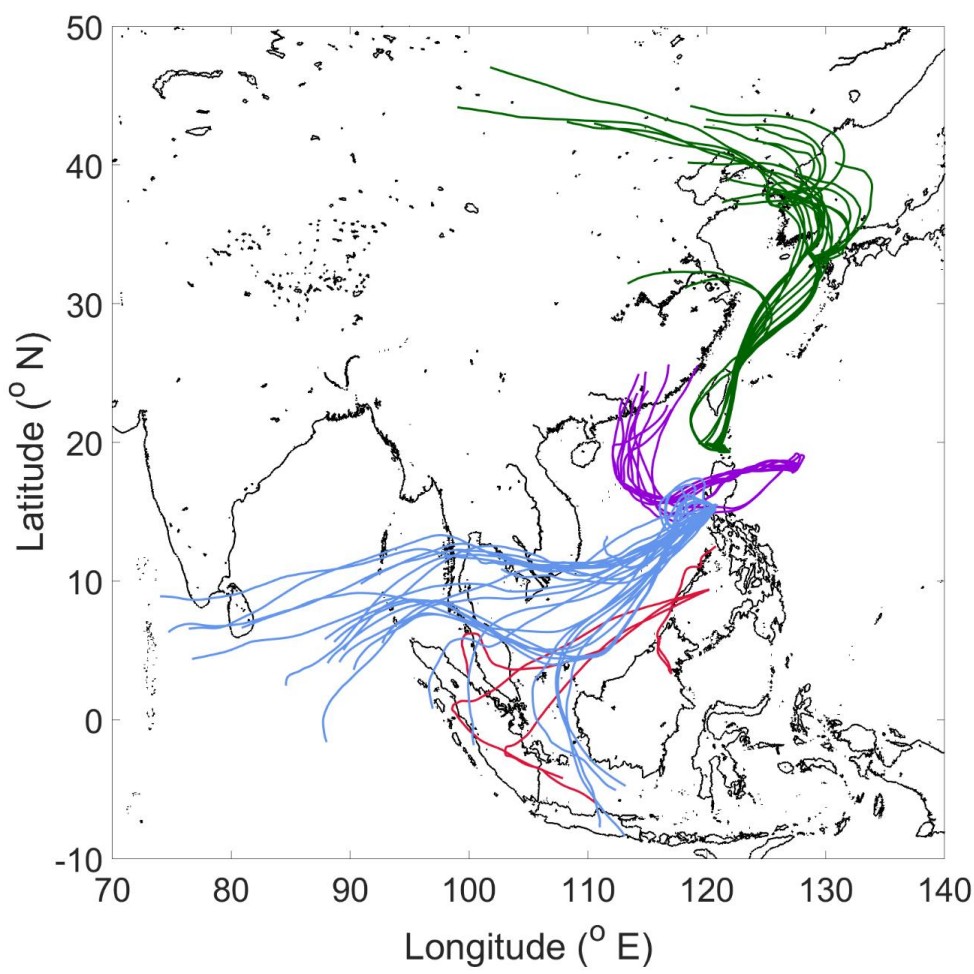

1123

**Figure 5:** Spatial summary of 120-hour back trajectories for each sample included in respective
case study sample sets: North (green; n = 20), East (purple; n = 11), Biomass Burning (red; n =
4), and Clark (blue; n = 25).

1127