# Peer review of "Total organic carbon and contribution from speciated organics in cloud water: Airborne data"

_Atmospheric Chemistry and Physics, 2021_

## Author Response (AR1)

Response: We thank the reviewers for thoughtful suggestions and constructive criticism that have helped us improve our manuscript. Below we provide responses to reviewer concerns and suggestions in blue font. All changes to the manuscript can be identified in the version submitted using Track Changes.

**Anonymous Referee #1:**

Stahl et al. report a unique and interesting set of cloud composition measurements in SE Asia. A large number of cloud water samples was collected from a research aircraft using a well-characterized axial-flow cloud sampler during the CAMP2Ex campaign, with analysis for inorganic ions and TOC as well as carboxylic and dicarboxylic acids, MSA, and dimethylamine. Observations of cloud water organic speciation are rare, especially from higher altitude clouds typically accessible only by aircraft, as are measurements more generally of cloud composition from this part of the world. The authors do a good job presenting this rich dataset and look at interesting phenomena such as variations in TOC with altitude and differences in TOC and organic speciation between clouds impacted or not impacted by biomass burning. They also do a good job placing their findings into the larger context available from other published studies. I do have several suggestions to improve the manuscript:

1. The authors need to think more about the effects of cloud water pH on their findings. Especially when studying uptake of gas phase weak acids, such as formic and acetic acids, cloud pH is a critical determinant of the effective Henry's Law solubility. Differences in the contributions of formate and acetate to measured TOC, a major focus of the manuscript, are quite possibly due to differences in pH between samples/regions. Because cloud water pH values in the region are likely roughly comparable to the pKa values for formic and acetic acids, even small changes in pH can lead to large changes in protonation/deprotonation state and significant resulting changes in partitioning from the gas phase. The manuscript mentions that cloud pH was measured. These measurements definitely need to be added to the paper. They are important in their own right but also critical to understanding gas-aqueous partitioning for key organic acids.

Response: The pH values have been added to Tables 1 and 2. The following text has been added to introduce the values:

Lines 349-351: "The pH of the cloud water with TOC measurements ranged from 3.79 – 5.93 and averaged 5.04 ± 0.51. The lowest pH values all occurred over the ocean."

2. Line 141: the statement here that the cloud sampler "efficiently collects cloud droplets with effective diameters > 20 um" is misleading and not especially helpful. The Crosbie et al. (2018) paper cited does talk about collecting a large fraction of cloud water when the cloud drop effective diameter is > 20 um, but this is not the most helpful comparison for the reader. As pointed out by Crosbie et al., the collector also efficiently collects smaller drops. The reference to "effective diameter" is not helpful here. The "effective diameter" is a property of the cloud drop size distribution. What is really relevant here, and what the authors should provide, is the collection efficiency of the cloud water collector as a function of physical cloud drop diameter. It is typical in such summary statements, for example, to report the 50% cut-size diameter of the collector.

Response: The airborne data collected so far with the AC3 has allowed only limited experimental characterization of the full collection efficiency curve. Despite efforts to use a Stokes number for velocity scaling, there appears to be some airborne platform dependence on the collection efficiency and therefore the determination of the 50% cut point; however, the airspeed of the P3 would be similar to the C130. Unlike laboratory characterization of particle collection systems (e.g. aerosol cyclones) where a monodisperse distribution of particles allows size dependence to be measured, the AC3 has been characterized, experimentally, using real clouds and in-flight conditions where the droplet size distribution is usually variable. The use of the effective diameter to characterize the size dependence was just a selection of one of many possible moments that could have been used by the authors. The shape of the size distribution and the variance (since there is an expected size dependence to collection) play a significant role in affecting the relationship between size and collection efficiency when attempting to utilize broad droplet size distribution data. With sufficient data across a dynamic range of conditions it will be possible to solve an inversion for the collection efficiency vs physical drop diameter (i.e., a monodisperse collection efficiency curve) and achieve what the reviewer is suggesting; however, that endeavor is currently ill-constrained.

3. Lines 145-146: The authors here point to a lack of leaching of organics from their plastic sample vials into the collected cloud water. More likely is the loss of cloud organics sticking to the surfaces of the plastic vials. Was this examined?

Response: That is an excellent point, laboratory tests we have done since getting this reviewer comment show that there is no appreciable difference in the peak areas indicating that there is no adsorption of organics to the walls of the conical vials. Additionally, any adsorption processes taking place would occur with the polypropylene IC vials, so any possible adsorption would be similar for all samples and would not qualitatively impact conclusions of this work. The following text has been added to clarify this point:

Lines 155-156: "Additional laboratory tests also indicated that there was no appreciable evidence that organics were adsorbing to the walls of the conical vials."

4. Line 159: How "immediate" was the analysis of collected cloud water? How much time elapsed between flying through a cloud and completing lab analysis? Loss of organic acids can be significant even at 24 hr after collection, due to rapid microbial degradation.

Response: The time between collecting and measuring samples is relatively variable depending on when in the flight they were collected. Samples collected during the flight were stored in a cooler with ice packs to keep them cold during the flight. Once the flight concluded, we would remove the cooler from the plane and take it directly to the lab for processing. Typically, it would be within 1-2 hours after the plane landed and, on average, each flight was about 7 hours. There is not much we could do about minimizing and holding fixed the storage time. When there was a back log of samples (due to back-to-back flights with large volumes of samples) the new samples were stored in IC vials and placed in a refrigerator until they could be queued up, which was typically less than 12 hours.

5. Line 200: acetic and formic acids "absorb" in cloud droplets; they do not so much "adsorb" to the droplet surface.

Response: Excellent point, the line now reads:

Line 218: "… acetic and formic acids absorb directly into droplets…"

6. The authors need to carefully evaluate their use of significant digits in the manuscript. The numbers of significant digits presented are often too large (normally one would present one digit more than the last digit reliably quantified) and inconsistent (e.g., 0.018 – 13.660 ppm C).

Response: The data values have been modified to include only four significant figures including in the text and tables.

7. Line 251: the Straub et al. samples were not collected in San Diego but over the Pacific ocean west of San Diego.

Response: Your comment is noted and the text now reads:

Line 271-272: "… lowest values being over the Pacific Ocean west of San Diego, California…"

8. Lines 267-268: differences between cloud water collectors in droplet sizes collected are relevant because cloud drop composition often varies across the cloud drop size spectrum. This point should be made more clearly for the reader and relevant references cited.

Response: Excellent point and the text now reads:

Line 290-291: "… droplet size, as well as compositional difference across the droplet size spectrum (i.e., Boris et al., 2016; Collett Jr. et al., 2008; Herckes et al., 2013)."

9. Line 270: Here and elsewhere, please state precisely what is meant by the +/- values given. Are these +/- one standard deviation?

Response: Noted, the addition "(± one standard deviation)" has been added to the following lines:

Lines 45, 293, 443, 487, 531, 596, 646

10. Lines 308-311: Why didn't the authors include H+ (from their pH measurements) in the calculated charge balances? For the weak organic acids, were the ionized fractions calculated, using sample pH, to properly estimate the charge balance? For example, part of the measured "acetate" concentration is present as acetate and part as acetic acid in the cloud sample.

Response: That is an excellent point and one which was overlooked in the initial calculation of the charge balance. An updated charge balance has been included in the supplemental. With the addition of both $H^+$ and $OH^-$ from the pH measurements, the charge balance slope shifted from 1.04 to 0.95. The following text has been updated to reflect the change:

Lines 330-334: "… 159 samples show an anion deficit (Fig. S1), with a slope of 0.95… This strong charge balance suggests… Species contributing to the anion deficit likely include a mix of unspeciated organic and inorganic anions.

11. Line 314: Na should be shown as Na+

Response: Noted, a superscripted "+" has been added to line 337.

12. The analysis of the BB-impacted periods is interesting, however, I am puzzled why the authors don't use the AMS m/z 60 smoke marker to support their analyses. At a minimum, the abundance of m/z 60 should be examined between periods identified as BB-impacted and those that are assumed to not be BB-impacted.

Response: That is an excellent point, and we did look into the use of m/z 60 and $f_{60}$ from the AMS. However, the m/z 60 data were sparse during in-cloud events with the CVI inlet in use. Additionally, there were very few out-of-cloud data points lining up with the 'before and after' cloud passes, which we thought was insufficient to draw any conclusions from.

13. The authors are generally cautious to not over-interpret correlations between species concentrations. To educate the reader, however, I suggest that they point out that correlations between cloud water species concentrations are sometimes high simply due to the common effect of LWC-related dilution across species.

Response: Noted, text has been added as follows:

Lines 221-224: "However, it should be noted that species concentrations in cloud water can be high simply due to the liquid water content being low, or inversely the concentrations can be low due to being diluted by high liquid water content."

14. Lines 483-488: The importance of cloud drop uptake of water-soluble organic gases should also be mentioned here.

Response: Noted, the following sentence has been added:

Lines 517-518: "However, the importance of droplet uptake of water-soluble organic gases should also be considered as they can influence TOC mass."

15. The use of acetate:formate ratios as a proxy for aged emissions is interesting, but the authors should demonstrate that these changes are not at least partly due to differences between the effective Henry's Law solubility of formic vs. acetic acids which will vary with cloud pH.

Response: This is an excellent point, however, based on the image below it can be seen that there is no clear relationship between pH value and either acetate or formate.

[Figure]

16. Lines 604-606: uptake of water-soluble organic gases can also be a factor contributing to greater organic mass contributions in cloud water, although this effect is also present for ammonia and nitric acid.

Response: Noted, the following lines have been added:

Lines 643-644: "The uptake of water-soluble gases can also attribute to greater organic mass contributions."

17. Lines 641-644: While organic acid adsorption onto coarse alkaline aerosols could well enhance uptake of these species in cloud water vs. their measurement in submicron aerosol sampled by the AMS, the alkaline nature of these coarse aerosols could also raise cloud pH and increase solubility of weak organic acids like formic and acetic acids. With the information available in this study, differentiating between these two effects is likely quite challenging.

Response: Yes, while that could be true, differentiating those two effects would be quite challenging as well as outside of the scope of our work. Indeed that would be an interesting analysis, however, even if it was within our scope, the current dataset has limitations and is missing critical information to accurately identify these alkaline dust particles (i.e., Al/Ti/Fe concentrations).

**Anonymous Referee #2:**

General Comments: This paper presents the chemical compositions of an impressive 159 cloud water samples, which were collected around the Philippines using a rare airborne technique during the CAMP2EX campaign. The concentrations of total organic carbon (TOC), organic acids, inorganic ions, and an alkyl amine were quantified, representing a large amount of effort in chemical analysis. While the collection and chemical analysis of these samples will be a substantial contribution to this field, the data analysis and interpretation require some additional work to reflect the strength of such a large and rare set of samples. I suggest that the authors: (1) further analyze and interpret the dataset to form a cohesive story; (2) re-evaluate and clarify the study objectives; then (3) reorganize to reflect the objectives.

The observations about the dataset presented in each subsection of the Results are generally clear, but it is difficult for the reader to deduce conclusions or a story. The majority of the Results section feels like a list of observations, rather than findings with interpretation and discussion. The section discussing Clark is more complete than the others; it would be ideal if the interpretation of other subsections of the Results were discussed in this way. More examples below.

The overall purpose and objectives of the paper seem to vary between sections, making the cohesive story of the paper unclear. In particular, the study hypotheses outlined in the Introduction section aim to contrast with recent work from Metro Manila; while this Stahl et al., 2020 aerosol study is clearly impactful, these hypotheses only frame a small part of your story (in addition, the contrast with this work is concerning because of a lack of control of variables; please see Specific Comments). While the contrast is interesting, either control over the variables should be established, or these hypotheses should be secondary within the paper. Of course there are many possibilities, but some suggestions of objectives based on the current Results section include: the presentation of a new set of samples collected using the AC3 cloud water collector or more generally at high altitude from this region; the contrast of the dataset with other atmospheric waters from the region and globally; the evidence for interactions between organic acids and dust/sea salt particles; the study of the cloud water vertical profiles of organic carbon and species.

Finally, to more clearly align with the paper results, the sections should be reorganized and revised to frame the same set of objectives. For example, the material in the first half of the Introduction is generally not explored in the Results. Also, the types of clouds from which samples were collected seems to be prominent in the abstract, but no further analysis by cloud type is discussed. Yet another example is that the Conclusion now includes insights that are not within the Results section (see Specific Comments).

Specific Comments

1. Some insights currently in the Conclusion would normally be placed in a Results and Discussion or Discussion section. The "Cumulative Results" section is somewhat confusing; I'm not certain where the discussion and interpretation was intended to go.

Response: We have doublechecked the Conclusion section and can confirm the current version does not include any new insight that was not provided in the main body of the paper. Without

specifics from the reviewer, it is hard to know what part of the Conclusion was a concern for them.

2.  There are some issues with comparing the results of the present cloud samples with the Stahl et al., 2020 aerosol samples from Metro Manila: (1) chemical components in the cloud droplets could also contain different transported material from other regions, and not specifically Metro Manila; (2) the samples were not collected concurrently, leading to possible different chemical sources and processes; and (3) some aerosol will remain interstitial in the cloud and not be observed in cloud chemistry.

Response: While it is true that the comparison to the Stahl et al. (2020) organic acid paper has some shortcomings as you have mentioned, it should also be mentioned that there is sparse data of this nature in the Philippines region. Therefore, broad comparisons and relationships can be drawn until a more precise study can properly link these comparisons.

3.  Figure 1: This figure is very similar to Figure 2 in Herckes, Valsaraj & Collett, 2013 with a few updates. At the least, an obvious mention that the figure is based on their Figure 2 is warranted, but please consider just referencing their figure instead of including it here. If you keep the figure, is there a particular order to the studies included? For example, in their Figure 2, the studies are organized by TOC concentration.

Response: A note has been made to signify that it is similar to Figure 2 in Herckes et al. (2013). The data is organized by continents (North America, South America, Africa, Europe, Asia) to compare regional measurements. Here is our added text:

Lines 1150-1151: "This figure is similar to that of Figure 2 in Herckes et al. (2013) with additional information presented and organized by continent."

4.  Section 2.3.1: How soon after were samples analyzed? Was this time fairly consistent between flights? Degradation of organic acids in atmospheric water samples without preservation from microbes and peroxides can affect the organic acid concentrations rapidly (e.g., for microbes: 10.1016/0960-1686(91)90198-G).

Response: Samples were prepared for the IC within 1-2 hours after being removed from the aircraft and this was a consistent procedure. In the event of back logged data (back-to-back flights with large quantities of samples), samples were stored in IC vials in the refrigerator until they could be queued up, which was usually less 12 hours after receiving the sample.

5.  The presented standard deviation values are confusing. In line 608, for example, the values appear to indicate that the contribution of these species to TOC mass in ~a third of the samples (based on a normal distribution) was below -5 %. Similar observations can be made about Table 3. Please clarify or check values. Is this simply a result of the data being non-normally distributed? Perhaps medians and percentiles should be used. Please state that the uncertainty values expressed throughout the paper are standard deviations.

Response: A clarification has been made stating that the "±" values were one standard deviation from the mean. While mathematically it is possible for negative percent concentrations to exist, physically it makes no sense and should be interpreted as zero percent (0%) for a lower bound.

6. Be careful about using specific organic acids as "markers" for sources. Organic acids are known to originate from many sources; using a particular species to demonstrate without uncertainty that an air mass containing it was from one of those sources is not often supported by the literature.

Response: While it is true that organic acids have many sources, we try to avoid definite classifications, but rather suggest they are from possible sources based on the data we have, such as back-trajectories or correlation with tracer species. Succinate has been documented to have strong association/correlation with biomass burning events and be an indicator that such an event is being sampled. However, this is not taken at face value and is corroborated with other sources such as back-trajectories or that the samples were collected in a known biomass burning plume.

7. The Four cases identified accounted for only 60 of the samples. The other 99 samples are not discussed in particular. Can anything be said about these "non-case" samples?

Response: The other 99 non-case samples were collected all around the Philippines across multiple flights, which made it hard to draw a proper story for those samples as there were too many variables involved (i.e., LWC, location, wind speed and direction, relative humidity) to perform a fair comparison. Cases such as the North, East, and Fire were all collected in a single flight, sequential, and in a relatively tight area. The only exception to this was the Clark case where samples over multiple flights were compiled; however, due to the similarity of the sample location and over land made it an optimal case for comparison purposes.

8. The implications of the DCA versus MCA comparisons, and the percent contributions to the subcategories, aren't clear since the acids analyzed differ from other studies. Why not discuss percent contributions to TOC or to total speciated organics (perhaps Table 3 could be percent of TOC)? Also, since glycolate is typically much less abundant than acetate and formate, I think it's more relevant to say that those two species were the most abundant, rather than saying that MCA concentrations were higher.

Response: While it is true that the percent contributions differ from other studies due to different species being measured, it seemed more appropriate to compare species based on what was measured. Additionally, percentages relative to measured DCA and MCA are more sensitive to events and can show compositional shifts more effectively. While acetate and formate dominate MCA concentrations, it is still relevant to include the other species to account for compositional shifts. It is expected for low molecular weight species such as acetate, formate, and oxalate to dominate their respective classifications, but shifts due to increased higher molecular weight compounds can signify important sources or processes, which is why we feel it is important to keep the classifications and subcategories as is.

9. Section 3.2 (Vertical Profiles): The analysis in this section in particular feels unfinished. There are many more questions you could answer with this dataset. What are the implications

of your findings? Are there any trends between specific species' vertical profiles? How do these profiles compare with past profiles of cloud water or even in-cloud aerosol (for example, 10.1002/2017JD027900, 10.1029/2012JD018089, or 10.5194/acp-20-3931-2020; the concentrations might need to be converted to air equivalent concentrations to compare)? Did you look at any other AMS ions or fractions of organics (e.g., f43, f44, or f60)?

Response: Yes, we do agree that many more questions could be answered, however, it could deviate from the scope of this article which is to focus on TOC (and associated species) concentrations. It should be noted that 159 samples are used for this analysis, however, a total of 304 cloud water samples were collected and speciated which would be more appropriate to use to get the full vertical profile story and compare variations such as cloud type, cloud phase, or small verse large droplet containing clouds. These additional cloud passes would give more opportunities for the use of the AMS organic fractions which we were unable to accommodate due to our limited samples that contain TOC measurements. For the purpose of this article we do not believe anything else needs to be added to the vertical profile analysis.

10. Please check significant digits throughout the paper; for example, some reported carboxylic acid concentrations have five significant digits, which is quite high, and in line 248 the TOC concentrations have many reported digits.

Response: The data has been trimmed to four significant figures in both the text and tables.

11. Can black carbon mass be removed from the Metro Manila aerosol to make the comparison of quantified organic/total mass reasonable between Metro Manila aerosol (Stahl et al., 2020) and the present work (for example, line 338, line 605)?

Response: The contribution of BC to the aerosol mass could be high for some stages but it did not have a major impact on the overall mass. In order to compare the influence of the measured organics to total mass, BC concentrations were removed from the total weight. Instead of a < 1% contribution of organics to the total mass, there is ~1.3% contribution to the total mass. It does not affect our conclusion; however, the text was updated to note the corrected computation:

Line 354-355: "… accounted for ~1.3% of total aerosol mass, excluding black carbon."

Line 640-641: "… compared to the surface layer aerosol measurements over Luzon, excluding black carbon (~1.3%)…"

12. DMA is the 4th most abundant speciated organic compound, but it was only observed above its LOD in one of the four cases. Please check that this makes sense.

Response: While that might be true, it should be noted that all measured species are above their respective LOD values, but then during calculations a background concentration is subtracted from the values, which can bring them down below the reported LOD values.

13. Why was malonate not quantified?

Response: Due to the IC instrument method used, malonate coeluted with carbonate making quantification difficult with large uncertainties.

14. Hilario et al., 2021 (10.5194/acp-21-3777-2021) appears to be quite related; that paper should be included in the discussion and interpretation of the present results.

Response: The Hilario et al. (2021) paper was considered and was used to classify cloud water samples based on the classified airmass types. However, upon analysis of the data and classifications, all but three samples were classified in the "Other" category and provided no use to our interpretation as a whole.

Technical Corrections

Please note: this is extensive; please do not respond to each of these.

1. Figure 3: Why are the species (organic acids, etc.,) presented in pie charts instead of vertical profiles? Is there information gained by presenting the more zoomed-in plots with zoomed-out overlays, or could (a), (c), and (d) be just the zoomed-out and give the same essential information? In (b), please label/describe which of the pies corresponds to which elevation.

Response: Plotting each organic species on a vertical profile would either: i) make the figure too cluttered if plotted on a single figure, or ii) there would be too many panels to be able to identify any of the characteristics. Therefore, it was determined to be better to plot the overall TOC versus altitude and include the compositional pie chart for each altitude "bin". While the zoomed-out figures show the same information as the zoomed-in figures, the data gets bunched together and makes it difficult to differentiate individual points. The pie charts correspond to the figure on the left and fall within each altitude "bin". The "bin" section lines will be extended into panel (b) to make it clear.

2. Figure 2: How was your map generated? Please clarify what is meant by the "midpoint of the cloud water samples" in the caption.

Response: The map was generated using coastline data and filled with a light gray background using MATLAB. Cloud water sampling start and stop times were averaged to get a center sampling point which was used to plot the sampling location for a given sample. Then each location was colored based on the TOC concentrations.

3. Please ensure that context is given for each new paragraph. For example, in line 531, "...absolute concentrations of most organics were greatly enhanced..." does not have a qualifier to specify which samples the statement refers to.

Response: The addition of "in BB" was added in line 562 to add context.

4. Parallel construction of plural nouns: for example, in line 534, "...glutarate ... and succinate ... accounted for a higher mass fraction than other cases...", there are two mass fractions discussed.

Response: The text has been modified to help make it clear.

Lines 565-568: "In the DCA population of species, glutarate and succinate accounted for higher mass fractions (17.15% and 41.95%) than other cases (0.65% – 4.02% and 20.82% – 38.52%, respectively).

5. Specify the quantity/parameter being discussed (statements should be literally correct): for example, in line 621, "...vertical profiles of AMS organic and m/z 44…", the quantity would be the mass or concentration of these fragments.

Response: Noted, the following has been added for clarity:

Line 662-663: "While vertical profiles of AMS organic and m/z 44 mass concentrations qualitatively resembled…."

6. Avoid the generic word "level" throughout your paper in favor of using a more precise word such as "concentration" or "mass".

Response: Noted, changes have been made throughout the text.

7. Line 84-85: if all types of fogs and clouds are being considered, 15 % is perhaps low. For example, see Figure 6 in https://doi.org/10.1016/j.atmosres.2013.06.005.

Response: We are specifically discussing organic acids, and while Figure 6 of Herckes et al. (2013) does show speciated contributions above 50%, but not all of those species are organic acids. It is true that some of the cases in Figure 6 show that even just the organic acids can account for more than 15% of the TOC, and this is true for some cases. However, as a generalization for both fog and cloud water based on other studies, we used the 15% threshold.

8. Line 106: Were the samples collected for this paper not part of the CAMP2EX campaign?

Response: The samples collected in this study were from the CAMP[2]Ex campaign. The other missions listed on lines 104-106 refer to other campaigns in the region that were not airborne based.

9. Line 109: While these airborne observations are clearly still important, it might be worth mentioning that there are some high elevation studies of organic acids in fog/cloud water that have been carried out in SE Asia. For example, Mount Tai in China (https://doi.org/10.5194/acp-17-9885-2017) and others you cite, Japan (https://doi.org/10.2343/geochemj.2.0601), and Jeju Island in Korea (https://doi.org/10.1016/j.atmosenv.2004.09.049; Decesari et al., 2005, which you cite).

Response: That is an excellent point. Additional text has been added to mention this.

Lines 109-111: "It should also be noted that there have also been a handful of high elevation studies carried out in Southeast Asia examining fog and cloud water organic acids (i.e., Decesari et al., 2005; Li et al., 2017; Mochizuki et al., 2020)."

10. Line 141: Based on the work of Crosbie et al., 2018, it seems that, "...efficiently collects cloud droplets with effective diameters $> 20$ μm" is a vast oversimplification. It would strengthen this paper if limitations of the sampling technique were directly addressed. This should include possible sampling concerns, especially droplet size-dependent collection efficiency (because chemical composition of droplets can depend on droplet size), evaporation of organic species, or concentration due to water evaporation. Please also add information about the setup of the AC3 during the campaign (and/or a relevant citation), such as pipe position. Correspondingly, in line 267 (Section 3.1), it would be beneficial to mention how the AC3 collection efficiency with respect to droplet size might be anticipated to affect differences in composition from other collectors.

Response: To address the comment about the oversimplified collection efficiency statement, the following lines have been added to the text to address the issue:

Lines 145-148: "The size dependence of the collection efficiency may influence the measured properties of the bulk cloud water in cases where there is a strong size-dependence in the droplet composition. Sample water evaporation was identified to affect low liquid water content environments and may increase aqueous concentrations."

The piping position was set at stage 10, which is described further in Crosbie et al. (2018), and was mounted on a fuselage pylon approximately 300 mm from the skin. The following text has been added:

Lines 148-150: "For this study the pipe position was set to position 10, as described in Crosbie et al. (2018), and mounted to the fuselage pylon approximately 300 mm from the skin."

As for the last comment we believe that no additional text is needed as it is purely speculative. The performance of other collectors is not being discussed and it would be hard to provide any supporting information to back up any speculation about whether another collector would "see" something different from these cloud samples.

11. Line 152: How was pH measured?

Response: The pH was measured using an Orion$^{TM}$ ROSS Ultra$^{TM}$ pH electrode with a precision of 0.01 using a two-point calibration at pH 4 and 7. The following text has been added to Section 2.3.2:

Lines 200-203: "The pH of the cloud water samples was measured using an Orion Star$^{TM}$ A211 pH meter with an Orion$^{TM}$ 8103BNUWP ROSS Ultra$^{TM}$ pH electrode (precision of 0.01). A two-point calibration (pH $= 4$ and pH $= 7$) was performed at the beginning of analyzing a particular flight's set of samples."

12. Line 152: please elaborate on the background removal. If it is accurate that the 10th percentile of all sample concentrations was subtracted as a background, please justify that procedure.

Response: This is a typo. The background is defined as the 10$^{th}$ percentile of the blanks, and not as the 10$^{th}$ percentile of the blanks plus the sample. The typo was corrected and a short sentence was added to clarify.

Lines 162-165: "A background was subtracted from the samples based on the bottom 10$^{th}$ percentile of all blanks collected during the campaign (both pre- and post-flight). The 10$^{th}$ percentile of the blanks was used instead of the mean as it is a compromise between removing the influence of background contamination and conserving data points."

13. Line 175: How were limits of detection calculated?

Response: Limits of detection were calculated for each species by using $3*S_ab^{-1}$ where $S_a$ is the standard deviation of the response and b is the slope of the calibration curve. The appropriate text has been added:

Lines 187-189: "… Table 1 and were calculated using $3S_ab^{-1}$ where $S_a$ is the standard deviation of the response and b is the slope of the calibration curve for that species."

14. Section 3.1, second paragraph: It would be more useful for the reader to focus this paragraph. Several of these species could be categorized as oxidation products mainly coming from aqueous reactions, and some could be particularly categorized as marine-sourced. Perhaps more importantly, some of the listed sources I find misleading. For example, I could not find any evidence of direct emission of oxalic acid/oxalate from biogenic sources in the Boone et al., 2015 paper cited, and oxalic acid has been discussed as being mainly from oxidation in the aqueous phase (10.1016/S1352-2310(03)00136-5, Warneck et al., 2003, is a clear example). In contrast, pyruvic acid has been found to be an oxidation product and to be directly emitted to the atmosphere (10.5194/acp-20-3697-2020). These should match interpretations in the Results.

Response: The idea of that paragraph is to give some background as well as possible sources and processes for the various organic acids. It is by no means a comprehensive list nor does it specify what our specific sources are, but rather gives the reader a broad idea of where these organics can come from. The Boone et al. (2015) paper was indeed a mis-cite and thank you for bringing it to our attention; the paper has been removed.

15. Line 309: Please use a more specific word than "good" and cite why. Please also list your source for expecting the deviation from 1.00 would be accounted for by H+ and metals (for example, the Straub 2017 paper). In addition, pH was measured; could the H+ be included in your calculations?

Response: Both H$^+$ and OH$^-$ have been added to the charge balance and the new figure can be seen in the supplemental. With the addition of H$^+$ the charge balance went from being cation

deficient to being anion deficient. The source for the missing anions is likely due to a mixture of unspeciated organics and inorganics. The text has been updated as follows:

Lines 330-334: "… 159 samples show an anion deficit (Fig. S1), with a slope of 0.95… This strong charge balance suggests… Species contributing to the anion deficit likely include a mix of unspeciated organic and inorganic anions.

16. Line 335: "...the measured ions in cloud water should contribute relatively more...." Please provide literature sources to support this. Please also consider that particle size is important in cloud nucleation and partitioning from the gas phase.

Response: References have been added:

Lines 360-365: "However, the measured ions in cloud water should contribute relatively more to total measured mass in cloud water owing to their hygroscopic nature (e.g., sea salt) and greater ease to become associated with cloud droplets as compared to more hydrophobic species (Chang et al., 2017; Dalirian et al., 2018; Pringle et al., 2010) like black carbon that contribute significantly to total aerosol mass in the boundary layer of Metro Manila (Cruz et al., 2019)."

17. Cases:

1. I would suggest putting North last so that the reader has more context. Please reorganize/revise the "North" section. In particular, the relationship to the Stahl, et al. organic acids concentrations and the split of the marine discussion are confusing.

2. What other studied air masses can be compared to these cases? What makes these samples unique or similar?

3. Please include more about the interpretation of the acetate/formate ratio as a marker for degree of oxidation. Most papers relate the ratio to biogenic versus anthropogenic sources, including in Talbot et al., 1988, Coggon et al., 2014 and elsewhere (10.1029/JD093iD02p01638). The Wang et al., 2007 paper is just one exception.

Response:

(1) We decided to start with North first because it had the highest sea salt influence, which in our view was a good starting point as sea salt is quite important with high global/marine relevance. The subsequent cases are more complicated as they include more influence from other sources. Re-arranging the sections could be viewed as a good idea based on who the reader is, but in our view the current order is fairly good.

(2) These cases were chosen because they were unique and it is hard to compare them meaningfully to other air masses. The cases are unique for the reasons described at the beginning of Section 4. We feel the opener of Section 4 lays the stage pretty well already for the ensuing case studies.

(3) Both Coggon et al. (2014) and Wang et al. (2007) discuss the use of the acetate-to-formate ratio as an indicator of direct emissions (fresh; high ratio) and photochemical processes

(secondary production; low ratio). Talbot et al. (1988) discussed the use of formic-to-acetic ratios where low ratios (high acetate-to-formate ratios) indicate direct emissions from motor vehicles or biomass burning (fresh) while high ratios (low acetate-to-formate ratios) were suggestive of biogenic processes, which could be in relation to secondary formation and photochemical processes. The following text has been added:

Lines 574-578: "Talbot et al. (1988) and Wang et al. (2007) both report that the acetate:formate ratio is substantially larger in biomass burning samples, which is contradictory to the ratios that are reported for this case ranging from 0.32 – 1.03. This could be due to the fuel type or due to aging of the biomass burning plume, however this is speculatory and should be examined more extensively."

18. In Section 4.3 (Biomass Burning), peat fires should be discussed because this is referenced in other papers throughout the paper.

Response: We felt it was unnecessary to discuss peat fires in detail as our primary focus of this article was not on biomass burning but on TOC concentrations. Biomass burning events were just specific cases to compare compositional and concentration differences from other samples. Any biomass burning specific article will likely discuss peat fire in greater detail, but we feel it unnecessary for this current work. Subsequent work can examine more of the details of peat fires as they relate to composition using the CAMP$^2$Ex dataset.

19. Line 502: Oxalate appears to be more abundant than succinate in biomass burning samples.

Response: Yes, oxalate is more abundant than succinate in biomass burning because oxalate is also emitted during the same process; however, oxalate also has other sources contributing to its overall concentration (e.g., including photo-oxidative decay of larger diacids like succinate) and so it is not surprising that it is higher than succinate. However, in comparison to the other cases, and even other samples, succinate concentrations are much higher owing to biomass burning being a major source of succinate.

20. Line 528: "This motivates more attention…": several studies, including those that you are citing, have done more in-depth chemical analysis of organic species in BB impacted cloud water.

Response: It is true that there are studies that have done more in-depth chemical analysis of BB impacted clouds, however, there is not much on high altitude or vertical profiles of these BB impacted clouds. To our knowledge the bulk of these types of measurements are predominately ground based and lack any vertical resolution.

21. Line 562: What implications does the observation of DMA have?

Response: The implications of DMA can include changes in hygroscopicity, possible aerosol nucleation events, and enhancements due to wildfire periods due to effective dissolution into cloud water (Youn et al., 2015). Because DMA is such a low contributor to mass we felt it was unnecessary to include this information.

22. Line 582: Why was adipate the only organic acid correlated with calcium ion if there is evidence of organic acids partitioning to dust particles?

Response: It is hard to conclude why this was the case without more reliable elements to trace to dust (i.e., Al/Ti); however, adipate is the longest organic acid we measured ($C_6$) which could have implications towards hydrophobic materials which could be associated with dust. Additionally, calcium is also associated with sea salt which could affect how the correlations are calculated. Non sea salt values could have been calculated and used, however, these can be erroneous as they are based off of the sodium concentrations which can also be emitted from crustal sources.

23. Line 614: Please state that the monocarboxylic acids measured were higher volatility than the dicarboxylic acids; monocarboxylic acids include less volatile species such as longer chain acids. Why would the precursors be gaseous?

Response: The text has been updated to add in your comment. Precursors are assumed to be gaseous due to the high concentration of likely gaseous species which could be formed from decomposition in the gas phase or in the aqueous phase. Gaseous precursors such as isoprene or monoterpene are examples but are just speculations due to lack of trace organic gases measurements.

Lines 655-658: "It should also be noted that MCAs have a higher volatility than DCAs, which could contribute to the higher organic mass. Additionally, the MCAs measured in this study were predominately short chain organics that have naturally higher volatilities (Chebbi and Carlier, 1996; Wang et al., 2007)."

24. Line 619: Is this usage of "significantly" accurate - does this indicate statistical significance?

Response: We changed the word to be "substantially".

25. Line 652: Succinate was not discussed previously as being from biomass burning.

Response: It was discussed in lines 546-548 that succinate and other organics have shown enhancements in concentrations during biomass burning events in the study region.

**References**

[revised manuscript text omitted]